# A Group-Theoretic Framework for Data Augmentation

**Shuxiao Chen**
Department of Statistics
University of Pennsylvania
shuxiaoc@wharton.upenn.edu

**Edgar Dobriban**
Department of Statistics
University of Pennsylvania
dobriban@wharton.upenn.edu

**Jane H. Lee**
Department of Computer Science
University of Pennsylvania
janehlee@sas.upenn.edu

## Abstract

Data augmentation has become an important part of modern deep learning pipelines and is typically needed to achieve state of the art performance for many learning tasks. It utilizes invariant transformations of the data, such as rotation, scale, and color shift, and the transformed images are added to the training set. However, these transformations are often chosen heuristically and a clear theoretical framework to explain the performance benefits of data augmentation is not available. In this paper, we develop such a framework to explain data augmentation as averaging over the orbits of the group that keeps the data distribution approximately invariant, and show that it leads to variance reduction. We study finite-sample and asymptotic empirical risk minimization and work out as examples the variance reduction in certain two-layer neural networks. We further propose a strategy to exploit the benefits of data augmentation for general learning tasks.

## 1 Introduction

Many deep learning models succeed by exploiting symmetry in data. Convolutional neural networks (CNNs) use that image identity is roughly invariant to translations: a translated cat is still a cat. Such invariances are present in many domains, including image and language data. Standard architectures are invariant to some, but not all transforms. CNNs induce an approximate equivariance to translations, but not to rotations. This is an *inductive bias* of CNNs, and the idea dates back at least to the neocognitron [30].

To make models invariant to arbitrary transforms beyond the ones built into the architecture, *data augmentation* (DA) is commonly used. The model is trained not just with the original data, but also with transformed data. Data augmentation is a crucial component of modern deep learning pipelines, and it has been used e.g., in AlexNet [44], and other pioneering works [17]. State-of-the-art results often rely on data augmentation. See Figure 1(a) for a small experiment.

Rather than designing new architectures, data augmentation is a universally applicable, generative, and algorithmic way to exploit invariances. However, a general theoretical framework for understanding augmentation is missing. Such a framework would enable us to reason clearly about its benefits and tradeoffs compared to invariant features. Further, we may discover the precise conditions under which data augmentation would lead to such benefits, which may guide its use in general learning tasks beyond domains in which it is standardly used.

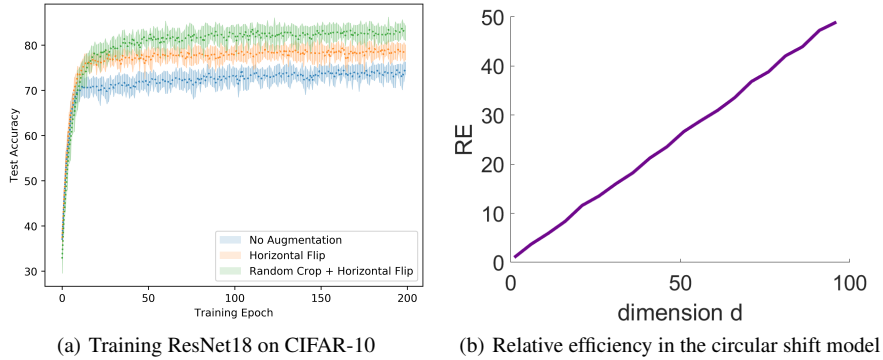

(a) Training ResNet18 on CIFAR-10

(b) Relative efficiency in the circular shift model

Figure 1: Benefits of data augmentation. Fig. (a) shows the test accuracy across training epochs of ResNet18 on CIFAR10 (1) without data augmentation, (2) horizontally flipping the image with 0.5 probability, and (3) randomly cropping a $32 \times 32$ portion of the image + random horizontal flip (See Appendix D for details). Fig. (b) shows the increase in relative efficiency achieved by data augmentation in an under-parameterized two-layer net under circular shift invariance (See Section 4.1 for details). Since relative efficiency is a ratio of MSE, this plot shows that using data augmentation improves MSE relative to not using augmentation, increasingly with higher input dimension.

In this paper, we propose such a general framework. We use group theory as a mathematical language, and model invariances as "approximate equality" in distribution under a group action. We propose that data augmentation can be viewed as invariant learning by averaging over the group action. We then demonstrate that data augmentation leads to sample efficient learning.

## 1.1 Our Contributions

**Novel theoretical framework and provable benefits.** We propose a novel probabilistic framework to model data augmentation with a group acting on the data so the distribution of the data does not change too much under the action (approximate invariance). Under this framework, we demonstrate that training with data augmentation is equivalent to learning with an *orbit-averaged* loss function, thus enabling us to prove a "universal" benefit of data augmentation in empirical risk minimization problems. To the best of our knowledge, there is essentially no prior work to show that data augmentation leads to provable statistical gains in generalization error, and even less to quantify precisely how much it benefits in a wide range of important settings.

**Invariance-variance tradeoff .** On a conceptual level, we characterize an intriguing relation between the size of the augmentation group and the performance gain. In Theorem 3.4 and 3.5, we show that the performance gain is governed by the superimposition of a variance reduction term and an additional bias term. While a larger group generally leads to greater variance reduction, it may lead to a larger bias. Thus, the best performance is achieved by delicately balancing the size of the group and the invariance, a phenomenon we termed as *invariance-variance tradeoff* .

**Precise formulas for variance reduction.** When the data distribution is exactly invariant under the group action, the additional bias term vanishes and we are able to obtain a precise formula for variance reduction, which depends on the covariance of the gradient along the group orbit (Theorem 4.1). This formula allows us to reason about the interplay between the network architecture and the choice of augmentation groups.

**Applications to deep learning and beyond.** We also work out examples of our theory for various learning tasks. We study the reduction in generalization error for over-parameterized two-layer nets (Theorem 3.6) using recent progress on Neural Tangent Kernels [38]. Moreover, we characterize the precise asymptotic gain in the *circular shift model*, an idealized two-layer net under circular shift invariance (Theorem 4.2). More examples, including exponential families, linear models, and certain non-linear parametric models, are deferred to Appendix B and C.

**Technical contributions.** Our general theory relies on a novel combination of optimal transport theory with tools in statistical learning theory, which may be of independent interest. Moreover, our results on the circular shift model are based on discrete Fourier analysis, which leads to an elegant

and perhaps surprising formula for the efficiency gain in terms of *tensor* and *Hadamard* products of the DFT matrix. For brevity, all proofs are deferred to Appendix A.

## 1.2 Prior Work

**Data augmentation methodology.** There is a lot of work in developing efficient methods for data augmentation. Related ideas date back at least to [6] (see also [59]). Today, state-of-the-art algorithms can automatically learn appropriate transformations for data augmentation with generative models [54, 32, 4, 62], with reinforcement learning [57, 23], with Bayesian methods [65], or with some carefully designed search schemes [24].

**Invariant architectures and representations.** Starting from [30, 45, 61, 14], a line of research focuses on designing (or learning) neural net architectures that are invariant to certain symmetries [31, 27, 70, 20, 22, 19], equivariant to SO(3) group [21, 28, 29], and others [58, 42, 69, 41]. Later, [12] gives a unified probabilistic treatment on designing invariant neural network. A related line of work focuses on representation learning and demonstrates the benefits of invariant feature representations [3, 50, 13, 1, 67, 53, 10, 2]. We remark that our goal is different compared on these two lines of work — while they focus on representation learning, we are concerned with the theoretical explanation of the benefits of data augmentation observed in practice.

**Prior theories on data augmentation.** To the best of our knowledge, prior theories on data augmentation are largely *qualitative*. For example, a line of work proposes to add random or adversarial noise when training [11, 51, 26, 74, 8, 49, 73, 37] and argue that this can lead to a form of regularization [63, 34, 35, 18, 71]. Other works have demonstrated connections between data augmentation and kernel classifiers [25], marginalized corrupted features [52], network architecture [9], margins [56], certain complexity measures [39], optimization landscape [48], and biological plausibility [36]. However, we have not found other works which explicitly prove a link between the data augmentation process and performance gain in the learning task.

## 2 Preliminaries and General Framework

We start in the setting of empirical risk minimization (ERM). Consider observations $X_1, \ldots, X_n \in \mathcal{X}$ (e.g., in supervised learning $X_i = (Z_i, Y_i)$ has both the features $Z_i$ and the label $Y_i$, such as images and classes) sampled i.i.d. from a probability distribution $\mathbb{P}$ on the sample space $\mathcal{X}$. Consider a group $G$ of transforms (e.g, the set of all rotations of images), which *acts* on the sample space: there is a function $\phi : G \times \mathcal{X} \to \mathcal{X}, (g, x) \mapsto \phi(g, x)$, such that $\phi(e, x) = x$ for the identify element $e \in G$, and $\phi(gh, x) =_d \phi(g, \phi(h, x))$ for $g, h \in G$. For notational simplicity, we write $\phi(g, x) \equiv gx$ when there is no ambiguity. To model invariance, we assume that for any $g \in G$, there is an "approximate equality" in distribution, made precise in Section 3:

$$X \approx_d gX, \qquad X \sim \mathbb{P}. \tag{1}$$

In supervised learning, $(Z, Y) \approx_d (gZ, Y)$ implies that given any class $Y = y$, e.g. a bird, the probability of observing an image $Z = z$ is close to that of observing the transformed image $gz$.

For a loss function $L(\theta, X)$, its empirical risk is defined as $R_n(\theta) := n^{-1} \sum_{i=1}^{n} L(\theta, X_i)$. We minimize $R_n(\theta)$ iteratively over time $t = 1, 2, \ldots$ using stochastic gradient descent (SGD) or variants. At each step $t$, a minibatch of $X_i$s (say with indices $S_t$) is chosen. In data augmentation, a random transform $g_{i,t} \in G$ is sampled and applied to each data point in the minibatch. Then, the parameter is updated via *augmented SGD* as

$$\theta_{t+1} = \theta_t - \frac{\eta_t}{|S_t|} \sum_{i \in S_t} \nabla L(\theta_t, g_{i,t} X_i). \tag{2}$$

We need a probability distribution $\mathbb{Q}$ on the group $G$, from which $g_{i,t}$ is sampled. For a finite $G$, one usually takes $\mathbb{Q}$ to be the uniform distribution. However, care must be taken if $G$ is infinite. We assume $G$ is a compact topological group, and we take $\mathbb{Q}$ to be the *Haar probability measure*[1]. Hence, for any $g \in G$ and measurable $S \subseteq G$, *translation invariance* holds: $\mathbb{Q}(gS) = \mathbb{Q}(S), \mathbb{Q}(Sg) = \mathbb{Q}(S)$.

A key observation is that the augmented SGD update rule corresponds to SGD on an *augmented empirical risk*, where we take an average over all augmentations according to the measure $\mathbb{Q}$:

$$\min_\theta \bar{R}_n(\theta) := \frac{1}{n} \sum_{i=1}^n \int_G L(\theta, gX_i) d\mathbb{Q}(g) = \frac{1}{n} \sum_{i=1}^n \bar{L}(\theta, X_i). \tag{3}$$

To be precise, $\nabla L(\theta, g_{i,t} X_i)$ is an unbiased stochastic gradient for the *augmented*, or *orbit-averaged* loss function $\bar{L}(\theta, X)$, and so we can view the resulting estimator as an empirical risk minimizer of $\bar{R}_n$. Hence, reasoning about the benefit of data augmentation is related to reasoning about the benefit of learning with the new loss function $\bar{L}$.

# 3 Data Augmentation Under Approximate Invariance

To deal with approximate invariance, we will leverage *optimal transport*, which provides a powerful framework to quantify closeness in distribution. Recall the notion of distance between probability distributions based on optimal transport (see, e.g., [68] for background and definitions needed):

**Definition 3.1** (Wasserstein metric). *Let $\mathcal{X}$ be a Polish space. Let $d$ be a lower semi-continuous metric on $\mathcal{X}$. For two probability distributions $\mu, \nu$ on $\mathcal{X}$, we define $\mathcal{W}_d(\mu, \nu) = \inf_{\pi \in \Pi(\mu,\nu)} \int_{\mathcal{X} \times \mathcal{X}} d(x, y) d\pi(x, y)$, where $\Pi(\mu, \nu)$ are all couplings whose marginals agree with $\mu$ and $\nu$. When $\mathcal{X}$ is a Euclidean space and $d$ is the Euclidean distance, we denote $\mathcal{W}_{\ell_2} \equiv \mathcal{W}_1$ and refer to it as the Wasserstein-1 distance.*

## 3.1 General Estimators and A Prototypical Invariance-Variance Tradeoff

Recall our goal is to characterize the performance of the empirical risk minimizer of the orbit-averaged loss $\bar{L}$. To lay the ground for this, we start by studying the effect of orbit-averaging for a generic measurable function $f$ of the data. Let $\bar{f}(x) = \mathbb{E}_{g \sim \mathbb{Q}} f(gx)$ be the orbit-averaged version of $f$. On the one hand, we expect some variance reduction by averaging a function over the orbit, and on the other hand, we expect a certain level of bias because the expectations of $f$ and $\bar{f}$ do not fully agree with each other. Specifically, we see a bias-variance tradeoff, made clear in the following lemma. Generally the subscripts to the operators $\mathbb{E}, \mathrm{Cov}$, such as $g, G, X$, denote the sources of randomness over which the expectations are evaluated.

**Lemma 3.2** (Approximate invariance lemma). *Let $f$ be s.t. each coordinate of the map $(X, g) \mapsto f(gX) \in \mathbb{R}^q$ is in $L^2(\mathbb{P} \times \mathbb{Q})$. Let $\bar{f}(x) := \mathbb{E}_{g \sim \mathbb{Q}} f(gx)$ be the "orbit average" of $f$. Let $\|f\|_\infty = \sup_x \|f(x)\|_2$ (which can be $\infty$). Then:*

1. *The expectations satisfy $\|\mathbb{E}_X \bar{f}(X) - \mathbb{E}_X f(X)\|_2 \le \mathbb{E}_g \mathcal{W}_1(f(gX), f(X))$;*
2. *The covariances satisfy $\mathrm{Cov}_X \bar{f}(X) = \mathrm{Cov}_{(X,g)} f(gX) - \mathbb{E}_X \mathrm{Cov}_g f(gX)$, and according to the Loewner order, we have[2]*

   $$\mathrm{Cov}_X \bar{f}(X) - \mathrm{Cov}_X f(X) \in [-\mathbb{E}_X \mathrm{Cov}_G f(gX) \pm 4\|f\|_\infty \cdot \mathbb{E}_g \mathcal{W}_1(f(gX), f(X)) \cdot I_q],$$

   *where $I_q$ is the identity matrix;*
3. *Let $\varphi$ be any real-valued convex function, and let $\overline{\varphi \circ f}(x) = \mathbb{E}_g \varphi \circ f(gx)$. Then*

   $$\mathbb{E}_X \varphi(\bar{f}(X)) - \mathbb{E}_X \varphi(f(X)) \in [(\mathbb{E}_X \varphi(\bar{f}(X)) - \mathbb{E}_X \overline{\varphi \circ f}(X)) \pm \|\varphi\|_{\mathrm{Lip}} \cdot \mathbb{E}_g \mathcal{W}_1(f(gX), f(X))],$$

   *where $\|\varphi\|_{\mathrm{Lip}}$ is the (possibly infinite) Lipschitz constant of $\varphi$.*

If we think of $f$ as an estimator of some functional of the data distribution, then $\bar{f}$ can be regarded as a data-augmented estimator, extending data augmentation beyond the ERM setup. In practice, $\bar{f}$ can be apporximated by Monte Carlo: sample $g_1, \ldots, g_k \sim \mathbb{Q}$, apply them to the data $X$, and then take the empirical average of $f(g_1 X), \ldots, f(g_k X)$. The following proposition characterizes the performance of this augmented estimator in terms of its mean-squared error (MSE):

**Proposition 3.3** (Benefits of data augmentation for general estimators). *Under the setup of Lemma 3.2, consider an estimator $\hat{\theta}(X)$ of some true population parameter $\theta_0$, and its augmented version $\hat{\theta}_G(X) = \mathbb{E}_{g \sim \mathbb{Q}} \hat{\theta}(gX)$. Then we have*

$$\mathrm{MSE}(\hat{\theta}_G) - \mathrm{MSE}(\hat{\theta}) \in [-\mathbb{E}_X \mathrm{tr}(\mathrm{Cov}_g \hat{\theta}(gX)) \pm \Delta], \tag{4}$$

where $\Delta = \mathbb{E}_g \mathcal{W}_1(\hat{\theta}(gX), \hat{\theta}(X)) \cdot [\mathbb{E}_g \mathcal{W}_1(\hat{\theta}(gX), \hat{\theta}(X)) + 2\|\text{Bias}(\hat{\theta}(X))\|_2 + 4\|\hat{\theta}\|_\infty]$.

The change in MSE (4) is the sum of two terms. From claim 2 of Lemma 3.2, $-\mathbb{E}_X \text{tr}\left(\text{Cov}_g \hat{\theta}(gX)\right)$ is the *variance reduction* $\text{Var}(\hat{\theta}_G) - \text{Var}(\hat{\theta})$ due to augmentation under exact invariance. The additional "bias" term $\Delta$ has three components: (1) the Wasserstein-1 distance $\mathbb{E}_g \mathcal{W}_1(\hat{\theta}(gX), \hat{\theta}(X))$ (which is small if the invariance is close to being exact), (2) the bias $\text{Bias}(\hat{\theta}(X))$ of the original estimator (which is small if the original estimator has small bias), and (3) the sup-norm $\|\hat{\theta}\|_\infty$ (which is small for bounded estimators). If the variance reduction is large, and the bias is small, then augmentation improves performance. This is a specific form of bias-variance tradeoff, which we refer to as the *invariance-variance tradeoff*.

The performance gain characterized by Proposition 3.3 is "universally applicable" to any estimator with finite variance. At this level of generality, we do not expect the bound (4) to be tight. In Appendix B, we present a tight analysis of $\hat{\theta}_G$ in linear models and show that sometimes this simple augmented estimator can achieve the same performance as the estimator obtained by constrained optimization over the "invariant parameter subspace". In general, there is reason to believe that constrained estimators can be more accurate but possibly harder to compute, and so it is remarkable that augmentation is equally accurate here.

We are now ready to present our results on augmented ERM. Let the population and empirical minimizers for the original risk and the augmented risk be

$$\theta_0 \in \arg\min_\theta \mathbb{E}L(\theta, X), \;\; \hat{\theta}_n \in \arg\min_\theta R_n(\theta), \;\; \theta_G \in \arg\min_\theta \mathbb{E}\bar{L}(\theta, X), \;\; \hat{\theta}_{nG} \in \arg\min_\theta \bar{R}_n(\theta),$$

where $\theta \in \Theta$ is in the parameter space. To systematically investigate the effect of learning with $\bar{L}$, we consider two evaluation criteria, namely the generalization error $\mathbb{E}L(\hat{\theta}, X) - \mathbb{E}L(\theta_0, X)$, and the parameter estimation error $\|\hat{\theta} - \theta_0\|_2$.

In the rest of this section, we show that the augmented estimator $\hat{\theta}_{nG}$ can outperform $\hat{\theta}_n$ based on both criteria, but because of two *fundamentally different reasons*. We will see in Section 3.2 that the reduction in generalization error can be quantified by *averaging the loss function* over the orbit, and in Section 3.3 that the reduction in parameter estimation error can be quantified by *averaging the gradient*. In both cases, we will see an invariance-variance tradeoff similar to that observed in Proposition 3.3.

## 3.2 Effect of Loss-Averaging

Classical theories on uniform concentration and Rademchaer complexity (see, e.g., [7, 60]) tell us that the generalization error of $\hat{\theta}_n$ can be quantified by how fast the empirical risk $R_n$ concentrates around its expectation, which can be further quantified by the Rademacher complexity of the loss class $\mathcal{R}_n(L \circ \Theta) := \mathbb{E} \sup_\theta |n^{-1} \sum_{i=1}^n \varepsilon_i L(\theta, X_i)|$ where $\varepsilon_i$'s are i.i.d. Rademacher random variables independent of $X_i$'s, and the expectation is taken over both $\{\varepsilon_i\}_1^n$ and $\{X_i\}_1^n$.

Following this intuition, we would expect that the generalization error of $\hat{\theta}_{nG}$ can also be quantified by how fast the concentration of the augmented risk $\bar{R}_n(\theta)$ happens. For one thing, because $\bar{R}_n(\theta)$ has additional averaging over $G$, we expect its concentration to happen *at a faster rate* than that of $R_n(\theta)$. But for another, the invariance-variance tradeoff kicks in because $\bar{R}_n$ concentrates around the wrong target $\mathbb{E}\bar{L}(\theta, X) \neq \mathbb{E}L(\theta, X)$ because of the non-exact invariance. This tradeoff is characterized by the following theorem:

**Theorem 3.4** (Effect of loss-averaging)**.** *Let $L(\theta, \cdot)$ be Lipschitz uniformly over $\theta$, with a (potentially infinite) Lipschitz constant $\|L\|_{\text{Lip}}$. Assume $L(\cdot, \cdot) \in [0, 1]$. Then with probability at least $1 - \delta$ over the draw of $X_1, \ldots, X_n$, we have*

$$\mathbb{E}L(\hat{\theta}_{n,G}, X) - \mathbb{E}L(\theta_0, X) \leq 2\mathcal{R}_n(\bar{L} \circ \Theta) + \sqrt{\frac{2\log(2/\delta)}{n}} + 2\|L\|_{\text{Lip}} \cdot \mathbb{E}_{g \sim \mathbb{Q}} \mathcal{W}_1(X, gX). \quad (5)$$

*Moreover, the Rademacher complexity of the augmented loss class can further be bounded as*

$$\mathcal{R}_n(\bar{L} \circ \Theta) - \mathcal{R}_n(L \circ \Theta) \leq \Delta + \|L\|_{\text{Lip}} \cdot \mathbb{E}_{g \sim \mathbb{Q}} \mathcal{W}_1(X, gX),$$

*where $\Delta = \mathbb{E} \sup_\theta |n^{-1} \sum_{i=1}^n \varepsilon_i \mathbb{E}_g L(\theta, gX_i)| - \mathbb{E}\mathbb{E}_g \sup_{\theta \in \Theta} |n^{-1} \varepsilon_i L(\theta, gX_i)| \leq 0$ is the "variance reduction" term.*

In contrast to (5), the classical Rademacher bound reads: $\mathbb{E}L(\hat{\theta}_n, X) - \mathbb{E}L(\theta_0, X) \leq 2\mathcal{R}_n(L \circ \Theta) + \sqrt{[2\log(2/\delta)]/n}$. Hence, the performance gain of $\hat{\theta}_{nG}$ is again governed by a variance reduction term $\Delta$, and an additional bias term $2\|L\|_{\text{Lip}} \cdot \mathbb{E}_g \mathcal{W}_d(X, gX)$, which vanishes under exact invariance. We remark that the inequality $\Delta \leq 0$ is based on an simple application of Jensen's inequality and can be loose. In practice, this term can be much smaller, so augmentation has a strong effect. A tight bound on it requires a case-by-case analysis, which we leave for future work.

### 3.3 Effect of Gradient-Averaging

In this subsection, we assume the sample space $\mathcal{X} \subseteq \mathbb{R}^d$ and the parameter space $\Theta \subseteq \mathbb{R}^p$. Classical theory on asymptotic statistics (see, e.g, [66]) tells that if both $d$ and $p$ are fixed and $n \to \infty$, $\hat{\theta}_n$ admits the following *Bahadur representation*: $\sqrt{n}(\hat{\theta}_n - \theta_0) = n^{-1/2}V_{\theta_0}^{-1}\sum_{i=1}^{n}\nabla L(\theta_0, X_i) + o_p(1)$, where $V_{\theta_0}$ is the hessian of $L$ at $\theta_0$. This is a first-order expansion of the estimator around the truth, implying convergence at the rate $n^{-1/2}$ and a central limit theorem. The validity of the Bahadur representation requires some standard assumptions, which we state below for completeness:

**Assumption A** (Regularity of the population risk minimizer). *The minimizer $\theta_0$ of the population risk is well separated: for any $\varepsilon > 0$, we have $\sup_{\theta:\|\theta - \theta_0\| \geq \varepsilon} \mathbb{E}L(\theta, X) > \mathbb{E}L(\theta_0, X)$.*

**Assumption B** (Regularity of the loss function). *For the loss function $L(\theta, x)$, we assume that*

1. *uniform weak law of large number holds: $\sup_\theta |\frac{1}{n}\sum_{i=1}^{n} L(\theta, X_i) - \mathbb{E}L(\theta, X)| \xrightarrow{p} 0$;*
2. *for each $\theta$, the map $x \mapsto L(\theta, x)$ is measurable;*
3. *the map $\theta \mapsto L(\theta, x)$ is differentiable at $\theta_0$ for almost every $x$;*
4. *there exists a $L^2(\mathbb{P})$ function $\dot{L}$ s.t. for almost every $x$ and for every $\theta_1, \theta_2$ in a neighborhood of $\theta_0$, we have $|L(\theta_1, x) - L(\theta_2, x)| \leq \dot{L}(x)\|\theta_1 - \theta_2\|$;*
5. *the map $\theta \mapsto \mathbb{E}L(\theta, X)$ admits a second-order Taylor expansion at $\theta_0$ with non-singular second derivative matrix $V_{\theta_0}$.*

It follows that $\hat{\theta}_n$ is asymptotically normal with covariance given by the inverse Fisher information, which gives an asymptotic characterization of the estimation error $\|\hat{\theta}_n - \theta_0\|$. Intuitively, we would expect $\hat{\theta}_{nG}$ also admits a Bahadur representation, with the gradient replaced by its orbit-averaged version. This is indeed correct under exact invariance (see Theorem 4.1), but care must be taken under approximate invariance.

**Theorem 3.5** (Effect of gradient-averaging). *Assume $\Theta$ is open and Assumptions A and B hold for both $(\theta_0, L)$ and $(\theta_G, \bar{L})$. In addition, for each $\theta$, assume the map $(X, g) \mapsto L(\theta, gX)$ is in $L^1(\mathbb{P} \times \mathbb{Q})$. Let $V_0, V_G$ be the Hessian of $\theta \mapsto \mathbb{E}L(\theta, X)$ and $\theta \mapsto \mathbb{E}\bar{L}(\theta, X)$ evaluated at $\theta_0$ and $\theta_G$, respectively. Let $M_0(X) = \nabla L(\theta_0, X)\nabla L(\theta_0, X)^\top$ and $M_G(X) = \nabla L(\theta_G, X)\nabla L(\theta_G, X)^\top$. Then we have*

$$\begin{aligned}
n(\text{MSE}(\hat{\theta}_{n,G}) - \text{MSE}(\hat{\theta}_n)) \to{}& n\|\theta_G - \theta_0\|_2^2 - \langle \mathbb{E}_X \text{Cov}_g(\nabla L(\theta_G, gX)), V_G^{-2}\rangle \\
&+ \mathbb{E}_{g,X}\langle M_G(gX) - M_G(X), V_G^{-2}\rangle + \mathbb{E}_X\langle M_G(X) - M_0(X), V_G^{-2}\rangle \\
&+ \langle \text{Cov}_X \nabla L(\theta_0, X), V_G^{-2} - V_0^{-2}\rangle
\end{aligned} \tag{6}$$

The MSE (6) again decreases due to a variance reduction term $\langle\mathbb{E}_X\text{Cov}_g(\nabla L(\theta_G, gX)), V_G^{-2}\rangle$ (which is non-negative since it is the inner product of two positive semi-definite matrices), but increases due to four additional bias terms. The first one, $n\|\theta_G - \theta_0\|^2$, comes from the fact that $\hat{\theta}_{nG}$ tends to $\theta_G$ (but not $\theta_0$) in the limit. The other three terms come from the fact that the empirical gradient and Hessian of the augmented loss function concentrate around the wrong target under approximate invariance, and thus can be regarded as the "first-order" and "second-order" bias induced by approximate invariance. Again, all of these additional bias terms vanish under exact invariance, illustrating the presence of the invariance-variance tradeoff.

### 3.4 A Case Study on Over-Parameterized Two-Layer Nets

In this subsection, we apply our general theory developed above to over-parameterized two-layer nets. Consider a binary classification problem, where the data points $\{X_i, Y_i\}_1^n \subseteq \mathbb{S}^{n-1} \times \{\pm 1\}$ are sampled i.i.d. from some data distribution. For technical convenience, we assume $|G| < \infty$, and $gX \in \mathbb{S}^{d-1}$ for all $g \in G$ and almost every $X$ from the feature distribution. Consider a two-layer net

$f(x; W, a) = m^{-1/2}a^\top\sigma(Wx)$, where $W \in \mathbb{R}^{m \times d}, a \in \mathbb{R}^m$ are the weights and $\sigma(x) = \max(x, 0)$ is the ReLU activation. We initialize the weights by $W_{ij} \sim \mathcal{N}(0, 1), a_s \sim \text{unif}(\{\pm 1\})$. Such a setup is common in recent literature on Neural Tangent Kernels (see, e.g., [38, 5, 40, 16]).

In the training process, we *fix $a$ and only train $W$*. We use the logistic loss $\ell(z) = \log(1 + e^{-z})$. In our previous notation, we have $L(\theta, X, Y) = \ell(Yf(X; W, a))$. The weight is then trained by gradient descent: $W_{t+1} = W_t - \eta_t\nabla\bar{R}_n(W_t)$, where $\bar{R}_n$ is the augmented empirical risk (3). To facilitate the analysis, we impose a margin condition below, similar to that in [40], which essentially says that there is a classifier that can distinguish the (augmented) data with positive margin.

**Assumption C** (Margin condition). *Let $\mathcal{H}$ be the space of functions $v : \mathbb{R}^d \to \mathbb{R}^d$ s.t. $\int\|v(z)\|_2^2 d\mu(z) < \infty$, where $\mu$ is the $d$-dimensional standard Gaussian probability measure. Assume there exists $\bar{v} \in \mathcal{H}$ and $\gamma > 0$, s.t. the Euclidean norm satisfies $\|\bar{v}(z)\|_2 \leq 1$ for any $z \in \mathbb{R}^d$, and that $Y\int\langle\bar{v}(z), gX\rangle\mathbf{1}\{\langle z, gX\rangle > 0\}d\mu(z) \geq \gamma$ for all $g \in G$ and amost all $(X, Y)$ from the data distribution.*

We need a few notations. For $\rho > 0$, we define $\mathscr{W}_\rho := \{W \in \mathbb{R}^{m \times d} : \|w_s - w_{s,0}\|_2 \leq \rho$ for any $s \in [m]\}$, where $w_s, w_{s,0}$ is the $s$-th row of $W, W_0$, respectively. We let $\mathcal{R}_n := \mathbb{E}\sup_{W \in \mathscr{W}_\rho}|n^{-1}\varepsilon_i[-\ell'(y_if(X_i; W, a))]|$ be the Rademacher complexity of the non-augmented gradient, where the expectation is taken over both $\{\varepsilon_i\}_1^n$ and $\{X_i, Y_i\}_1^n$. Similarly, writing $f_{i,g}(W) = f(gX_i; W, a)$, we define $\bar{\mathcal{R}}_n := \mathbb{E}\sup_{W \in \mathscr{W}_\rho}|n^{-1}\varepsilon_i\mathbb{E}_g[-\ell'(Y_if_{i,g}(W))]|$ to be the Rademacher complexity of the augmented gradient. The following theorem characterizes the performance gain by data augmentation in this example:

**Theorem 3.6** (Benefits of data augmentation for overparameterized two-layer nets). *Under Assumption C, take any $\varepsilon \in (0, 1)$ and $\delta \in (0, 1/5)$. Let*

$$\lambda = \frac{\sqrt{2\log(4n|G|/\delta)} + \log(4/\varepsilon)}{\gamma/4}, \quad M = \frac{4096\lambda^2}{\gamma^6}, \quad \rho = 4\lambda/(\gamma\sqrt{m}).$$

*Let $k$ be the best iteration (with the lowest empirical risk) in the first $\lceil 2\lambda^2/n\varepsilon\rceil$ steps. Let $\alpha = 16[\sqrt{2\log(4n|G|/\delta)} + \log(4/\varepsilon)]/\gamma^2 + \sqrt{md} + \sqrt{2\log(1/\delta)}$. For any $m \geq M$ and any constant step size $\eta \leq 1$, with probability at least $1 - 5\delta$ over the random initialization and i.i.d. draws of the data points, we have*

$$\mathbb{P}(Yf(X; W_k, a) \leq 0) \leq 2\varepsilon + [\sqrt{\frac{2\log(2/\delta)}{n}} + 4\bar{\mathcal{R}}_n] + \frac{1}{2}\mathbb{E}_Y\mathbb{E}_g\mathcal{W}_1(X|Y, gX|Y)\cdot\alpha. \quad (7)$$

*The three terms bound the* optimization error, generalization error, *and the* bias due to approximate invariance. *Moreover, with probability at least $1 - \delta$ over the random initialization, we have*

$$\bar{\mathcal{R}}_n - \mathcal{R}_n \leq \Delta + \frac{1}{4}\mathbb{E}_Y\mathbb{E}_g\mathcal{W}_1(X|Y, gX|Y)\cdot\alpha, \quad (8)$$

*where $\Delta = \mathbb{E}\sup_{W \in \mathscr{W}_\rho}|n^{-1}\varepsilon_i\mathbb{E}_g[-\ell'(y_if_{i,g}(W))]| - \mathbb{E}\mathbb{E}_g\sup_{W \in \mathscr{W}_\rho}|n^{-1}\varepsilon_i[-\ell'(y_if_{i,g}(W))]| \leq 0$ is the "variance reduction" term.*

The above theorem resembles Theorem 2.2 in [40] — we use a corollary of their Theorem 2.2 to show the optimization error can be made arbitrarily small. However, the treatment of generalization error is non-trivial and is based on our technical tools developed in the previous subsections. Specifically, we need to decompose the generalization error into two terms: one of them is dealt with using uniform concentration and Rademacher complexity (but with McDiarmid's inequality replaced by Talagrand's concentration inequality), and the other one is handled by exploiting the distributional closeness between $(gX, Y)$ and $(X, Y)$ with tools from optimal transport theory, similar to the strategy used in the proof of Theorem 3.4 and 3.5. Again, we see an invariance-variance tradeoff in this example due to approximate invariance.

## 4 Data Augmentation Under Exact Invariance

When the invariance is exact, so $X =_d gX$ for almost all $g \in G$ and $X \sim \mathbb{P}$, all bias terms in the previous results vanish, and we are left with a variance reduction term. In this section, we present a formula for this term which is asymptotically exact and leads to additional insights.

**Theorem 4.1** (Asymptotic formula for variance reduction). *Assume $\Theta$ is open and let the pair $(\theta_0, L)$ satisfy Assumptions A and B. In addition, for each $\theta$, assume the map $(X, g) \mapsto L(\theta, gX)$ is in $L^1(\mathbb{P} \times \mathbb{Q})$. Then under exact invariance, we have*

$$\sqrt{n}(\hat{\theta}_n - \theta_0) \Rightarrow \mathcal{N}(0, \Sigma_0), \quad \sqrt{n}(\hat{\theta}_{nG} - \theta_0) \Rightarrow \mathcal{N}(0, \Sigma_G), \tag{9}$$

*where the classical and augmented covariances are*

$$\Sigma_0 = V_0^{-1} \mathbb{E}_X[\nabla L(\theta_0, X) \nabla L(\theta_0, X)^\top] V_0^{-1}, \ \Sigma_G = \Sigma_0 - V_0^{-1} \mathbb{E}_X[\mathrm{Cov}_g \nabla L(\theta_0, gX)] V_0^{-1}, \tag{10}$$

*and we recall that $V_0$ is the Hessian of $\theta \mapsto \mathbb{E}_X L(\theta, X)$ evaluated at $\theta_0$. As a consequence the $\hat{\theta}_{nG}$ gains efficiency compared to $\hat{\theta}_n$, and has a relative efficiency of $\mathrm{RE} = \mathrm{tr}(\Sigma_0)/\mathrm{tr}(\Sigma_G) \geq 1$.*

Note that different from Theorem 3.5, we only impose assumptions on the pair $(\theta_0, L)$. This makes the proof non-trivial and we need a careful limiting argument to justify the asymptotic normality of $\hat{\theta}_{nG}$.

This result rigorously shows that data augmentation helps if $\mathrm{tr}\, V_{\theta_0}^{-1} \mathbb{E}_X[\mathrm{Cov}_g \nabla L(\theta_0, gX)] V_{\theta_0}^{-1} > 0$. It gives a precise formula for the asymptotic variance reduction. The term $\mathbb{E}_X \mathrm{Cov}_g \nabla L(\theta_0, gX)$ is the average covariance of the gradient $\nabla L$ along the orbits $Gx$, which is large if the gradient varies a lot along the orbits. This is consistent with the intuition that one should choose $G$ to encode symmetries *not captured* by the network architecture. Indeed, if the neural network is already invariant to actions by $G$, the covariance along the orbit is zero, meaning no performance gain by data augmentation.

## 4.1 The Circular Shift Model

In this subsection, we apply Theorem 4.1 to give an even more explicit formula for the variance reduction term in the *circular shift model*, which is essentially an under-parameterized two-layer neural network under circular shift invariance.

Consider a regression problem, where we observe an i.i.d. random sample $\{(X_i, Y_i)\}_1^n \subseteq \mathbb{R}^d \times \mathbb{R}$ distributed as the a random vector $(X, Y)$. We assume the data is generated from the following two-layer neural network model $Y = a^\top \sigma(WX) + \varepsilon, \varepsilon \perp X, \mathbb{E}\varepsilon = 0$, where $X$ is a $d$-dimensional input, $W \in \mathbb{R}^{m \times d}, a \in \mathbb{R}^m$ are the weights, and $\sigma$ is the activation function. In this subsection, for simplicity, we assume $a = 1_m$ is the all-one vector, and $\sigma(x) = x^2$.

We have a group $G$ acting on $\mathbb{R}^d \times \mathbb{R}$ *only through $X$*: $g(X, Y) = (gX, Y)$. The invariance is characterized by $(gX, Y) =_d (X, Y)$. Here we focus on a natural example of *circular shift invariance*, where $G = \{g_0, g_1, \ldots, g_{m-1}\}$, and $g_i$ acts by shifting a vector *circularly* by $i$ units: $(g_i x)_{j+i \bmod m} = x_j$.

We will use the square loss: $L(\theta, X, Y) = (Y - f(\theta, X))^2$. We assume the optimization is successful, so that $\hat{W}_n$ and $\hat{W}_{nG}$ are indeed global minimizers of the non-augmented and augmented empirical risks, respectively. The following theorem characterizes the efficiency gain of data augmentation for fixed $m, d$ and $n \to \infty$. It uses some notions from discrete Fourier analysis, specifically the discrete Fourier transform (DFT) matrix $F$, the $d \times d$ matrix defined by $F_{j,k} = d^{-1/2} \omega^{jk}$, where $\omega$ is the $n$-th root of unity.

**Theorem 4.2** (Variance reduction formula in the circular shift model). *Consider the circular shift model. Assume the conditions in Theorem 4.1 hold, and the population risk $\mathbb{E}L(\cdot, X, Y)$ is twice differentiable. Let $\mathbb{E}\varepsilon^2 = \gamma^2$. Define $C_v$ to be the circulant matrix associated with the vector $v$, whose $(i, j)$-th entry is $v_{i-j+1}$. Then*

$$\sqrt{n}(\hat{W}_n - W) \Rightarrow \mathcal{N}(0, \gamma^2 I_W^{-1}), \qquad \sqrt{n}(\hat{W}_{n,G} - W) \Rightarrow \mathcal{N}(0, \gamma^2 I_W \bar{I}_W^{-1} I_W), \tag{11}$$

*where*

$$I_W = (W \otimes W) \cdot \mathbb{E}[XX^\top \otimes XX^\top], \qquad \bar{I}_W = (W \otimes W) \cdot d^{-2} \mathbb{E}[C_X C_X^\top \otimes C_X C_X^\top]. \tag{12}$$

*If the features are normally distributed $X \sim \mathcal{N}(0, I_d)$, then*

$$I_W = (W \otimes W) \cdot S, \qquad \bar{I}_W = (W \otimes W) \cdot F_2^* \cdot (F_2^2 \odot M) \cdot F_2^*, \tag{13}$$

*where $S$ is the $d \times d \times d \times d$ tensor whose $(i, j, i', j')$-th entry is 3 if $i = j = j' = j'$, is 1 if there are two distinct indices among $(i, j, i', j')$, and is zero otherwise, $F_2 = F \otimes F$, $F_2^*$ is its complex conjugate, and $M$ is the $d \times d \times d \times d$ tensor whose $(i, j, i', j')$-th entry is $F_i^\top F_j \cdot F_{i'}^\top F_{j'} + F_i^\top F_{j'} \cdot F_{i'}^\top F_j + F_i^\top F_{i'} \cdot F_i^\top F_{j'}$.*

In (13), we discover elegant and perhaps surprising formula for the augmented Fisher information $\bar{I}_W$ in terms of *tensor* ($\otimes$) and *Hadamard* ($\odot$) products of the DFT matrix.

To get a sense of these formulas, we study how much "smaller" $\bar{I}_W$ is compared to $I_W$ by calculating the average MSEs, i.e., the averages of their traces. First, we have $\mathbb{E}\operatorname{tr} I_W = \mathbb{E}\|WXX^\top\|_F^2$. If $X \sim \mathcal{N}(0, I_d)$, then this quantity is equal to $p \operatorname{tr} S^2 = p \operatorname{tr}(XX^\top)^2$. Similarly, one can calculate that $\mathbb{E}\operatorname{tr} \bar{I}_W = p \operatorname{tr}(C_X C_X^\top)^2/d^2$. In Figure 1(b), we show the results of an experiment where we randomly generate the input as $X \sim \mathcal{N}(0, I_d)$. We compute the values of $\mathbb{E}\operatorname{tr} I_W = p \operatorname{tr}(XX^\top)^2$ and $\mathbb{E}\operatorname{tr} \bar{I}_W = p \operatorname{tr}(C_X C_X^\top)^2/d^2$, and record their ratio. We repeat the experiment $n_{MC} = 100$ times. We then show the relative efficiency as a function of the input dimension $d$. We find that the relative efficiency scales as $RE(d) \sim d/2$. Thus, the efficiency gain increases as a function of the input dimension. However, the efficiency gain does not depend on the output dimension $m$. This makes sense, as circular invariance reduces only the input dimension.

## 5  Discussion

To summarize, we propose a novel probabilistic framework for data augmentation, under which training with augmented data is equivalent to learning with an orbit-averaged loss. Based on this observation, we rigorously prove data augmentation reduces the generalization error as well as the estimation error in ERM problems. Section 3 introduces a general framework to model the data augmentation process, showing the existence of an invariance-variance tradeoff when $G$ keeps the data distribution only approximately invariant. Section 4 considers exact invariance to further explore the variance reduction term, and show there are more gains when the gradient of the loss varies a lot along the orbits. In both sections, we provide concrete examples (and more in the Appendix) of using our framework to derive more explicit formulas for variance reduction specific to a given setup.

The current paper only deals with the vanilla "label-preserving" data augmentation with a uniform sampling scheme (i.e., the measure we put on $G$ is Haar). Note that all of our current results would hold if $G$ is only a semi-group, provided we can endow it with a uniform probability measure. We would encourage further work to extend our results to other types of augmentations such as the *mixup* [73] and to more general sampling schemes. We would also encourage future work to investigate the invariance-variance tradeoff in more general setups, as this tradeoff is essential in choosing the augmentation groups in practice. Furthermore, we believe these results have potential for suggesting ways to judge the quality of an augmentation without actual training, but this is not explicitly explored in this paper. We believe this topic deserves its own treatment in future work, and encourage exploration in this direction.

## Broader Impact

Our probabilistic framework is to our knowledge the first of its kind to rigorously prove and quantify how data augmentation helps the learning task. Further, the invariance-variance tradeoff may provide new insight for how we think about choosing and composing transformations used in data augmentation. Since data augmentation is routinely used in modern deep learning pipelines and it is emerging in other applications, like self-supervised learning (see, e.g., [15]), our theoretical framework developed in this paper could potentially be used to provide guidance on the choice of the augmentation groups in these application domains.

However, care must be taken when using our theories to develop new augmentation strategies. We've demonstrated that the benefits of data augmentation crucially depends on the invariance structures present in the data distribution — if one chooses an augmentation group which does not correctly capture the invariance structures, or if the data at hand simply has no such structures, then blindly using data augmentation can potentially harm the model performance. In this sense, domain expertise is required to fully harness the power of our theories.

## Acknowledgments and Disclosure of Funding

This work was supported in part by NSF BIGDATA grant IIS 1837992 and NSF TRIPODS award 1934960. We thank the Simons Institute for the Theory of Computing for providing AWS credits.

## Footnotes

[1]Haar measures are used for convenience. Most of our results hold for more general measures with slightly more lengthier proofs.

[2]For symmetric matrices $A, B, C$, we will use the notation $A \in [B, C]$ to mean that $B \preceq A \preceq C$ in the Loewner order.

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
