[Supplementary Material]

# A Omitted Proofs

## A.1 Proof of Lemma 3.2

For part 1, we have

$$\|\mathbb{E}_X \mathbb{E}_g f(gX) - \mathbb{E}_X f(X)\|_2 = \sup_{\|v\|_2 \leq 1} \mathbb{E}_g \mathbb{E}_X \langle v, f(gX) - \mathbb{E}_X f(X) \rangle$$
$$\leq \sup_{\|v\|_2 \leq 1} \mathbb{E}_g \|v\|_2 \mathcal{W}_1(f(gX), f(X))$$
$$= \mathbb{E}_g \mathcal{W}_1(f(gX), f(X)),$$

where the inequality is due to Kantorovich-Rubinstein theorem, i.e., the dual representation of the $W_1$ metric (see. e.g., [68]).

For part 2, by law of total variance, we have

$$\text{Cov}_X \bar{f}(X) - \text{Cov}_X f(X) = -\mathbb{E}_X \text{Cov}_g f(gX) + \Delta_1 + \Delta_2,$$

where

$$\Delta_1 = \mathbb{E}_{(X,g)} f(gX) f(gX)^\top - \mathbb{E}_X f(X) f(X)^\top$$
$$\Delta_2 = \mathbb{E}_X f(X) \mathbb{E}_X f(X)^\top - \mathbb{E}_{(X,g)} f(gX) \mathbb{E}_{(X,g)} f(gX)^\top$$

For any non-zero vector $v$, we have

$$|v^\top \Delta_1 v| = \left| \mathbb{E}_g \mathbb{E}_X \left[ \langle v, f(gX) \rangle^2 - \langle v, f(X) \rangle^2 \right] \right|$$
$$\leq \mathbb{E}_g \left| \mathbb{E}_X \left[ \langle v, f(gX) \rangle^2 - \langle v, f(X) \rangle^2 \right] \right|.$$

The Lipschitz constant for the function $w \mapsto \langle v, w \rangle^2$, where $w \in \text{Range}(f)$, is bounded above by $2\|v\|_2^2 \|f\|_\infty$. Invoking Kantorovich-Rubinstein theorem again, we have

$$|v^\top \Delta_1 v| \leq 2\|v\|_2^2 \|f\|_\infty \mathbb{E}_g \mathcal{W}_1(f(gX), f(X)).$$

Similarly, we have

$$|v^\top \Delta_2 v| = \left| \left( \mathbb{E}_X \langle v, f(X) \rangle \right)^2 - \left( \mathbb{E}_{(X,g)} \langle v, f(gX) \rangle \right)^2 \right|$$
$$\leq 2\|f\|_\infty \left| \mathbb{E}_g \mathbb{E}_X \left[ \langle v, f(X) \rangle - \langle v, f(gX) \rangle \right] \right|$$
$$\leq 2\|f\|_\infty \mathbb{E}_g \left| \mathbb{E}_X \left[ \langle v, f(X) \rangle - \langle v, f(gX) \rangle \right] \right|$$
$$\leq 2\|v\|_2^2 \|f\|_\infty \mathbb{E}_g \mathcal{W}_1(f(gX), f(X)).$$

Part 2 is then proved by recalling the definition of the Loewner order.

For part 3, we have

$$\mathbb{E}_X \varphi(\bar{f}(X)) - \mathbb{E}_X \varphi(f(X)) = \mathbb{E}_X \varphi(\bar{f}(X)) - \mathbb{E}_X \varphi \circ \bar{f}(X) + \mathbb{E}_g \mathbb{E}_X \varphi \circ f(gX) - \mathbb{E}_X \varphi(f(X)).$$

We finish the proof by noting that

$$\left| \mathbb{E}_g \mathbb{E}_X \varphi \circ f(gX) - \mathbb{E}_X \varphi(f(X)) \right| \leq \|\varphi\|_{\text{Lip}} \mathcal{W}_1(f(gX), f(X)).$$

## A.2 Proof of Proposition 3.3

For notational simplicity, we let $\bar{f} = \hat{\theta}_G$ and $f = \hat{\theta}$. Then using bias-variance decomposition, we have

$$\text{MSE}(\bar{f}) - \text{MSE}(f) = B + V,$$

where

$$B = \|\text{Bias}(\bar{f})\|_2^2 - \|\text{Bias}(f)\|_2^2$$
$$V = \text{tr}(\text{Cov}_X \bar{f}(X)) - \text{tr}(\text{Cov}_X f(X)).$$

We first analyze the bias term. Note that by Lemma 4.2, we have

$$\left| \|\text{Bias}(\bar{f})\|_2 - \|\text{Bias}(f)\|_2 \right| \leq \|\mathbb{E}_X \bar{f}(X) - \mathbb{E}_X f(X)\|_2$$
$$\leq \mathbb{E}_g \mathcal{W}_1(f(gX), f(X)).$$

Hence

$$|B| \leq \left( \|\text{Bias}(\bar{f})\|_2 + \|\text{Bias}(f)\|_2 \right) \left| \|\text{Bias}(\bar{f})\|_2 - \|\text{Bias}(f)\|_2 \right|$$

$$\leq \left( \|\mathbb{E}_X \bar{f}(X) - \mathbb{E}_X f(X)\|_2 + 2\|\text{Bias}(f)\|_2 \right) \cdot \|\mathbb{E}_X \bar{f}(X) - \mathbb{E}_X f(X)\|_2$$

$$\leq \left( \mathbb{E}_G \mathcal{W}_1(f(gX), f(X)) + 2\|\text{Bias}(f)\|_2 \right) \cdot \mathbb{E}_g \mathcal{W}_1(f(gX), f(X)).$$

The variance term $V$ can be bounded by the following arguments. We have

$$|\text{tr}(\Delta_1)| = \left| \mathbb{E}_g \mathbb{E}_X \left[ \|f(gX)\|_2^2 - \|f(X)\|_2^2 \right] \right|$$

$$\leq 2\|f\|_\infty \mathbb{E}_g \left| \mathbb{E}_X \left[ \|f(gX)\|_2 - \|f(X)\|_2 \right] \right|$$

$$\leq 2\|f\|_\infty \mathbb{E}_g \mathcal{W}_1(f(gX), f(X)).$$

Similarly, we have

$$|\text{tr}(\Delta_2)| = \left| \|\mathbb{E}_X f(X)\|_2^2 - \|\mathbb{E}_{X,g} f(gX)\|_2^2 \right|$$

$$\leq 2\|f\|_\infty \left| \|\mathbb{E}_X f(X)\|_2 - \|\mathbb{E}_{X,g} f(gX)\|_2 \right|$$

$$\leq 2\|f\|_\infty \|\mathbb{E}_X f(X) - \mathbb{E}_X \bar{f}(X)\|_2$$
$$\leq 2\|f\|_\infty \mathbb{E}_g \mathcal{W}_1(f(gX), f(X)),$$

where the last inequality is due to Lemma 3.2.

Combining the bound for $B$ and $V$ gives the desired result.

### A.3 Proof of Theorem 3.4

We first prove a useful lemma.

**Lemma A.1** (Triangle inequality/Tensorization). *For two random vectors $(X_1, ..., X_n)$, $(Y_1, ..., Y_n) \in \mathcal{X}^n$, we denote the joint laws as $\mu^n, \nu^n$ respectively, and the marginal laws as $\{\mu_i\}_1^n, \{\nu_i\}_1^n$ respectively. We have*

$$\mathcal{W}_{d^n}(\mu^n, \nu^n) \leq \sum_i \mathcal{W}_d(\mu_i, \nu_i).$$

*Proof.* By Kantorovich duality (see, e.g., [68]), for each coordinate, we can choose optimal couplings $(X_i^*, Y_i^*) \in \Pi(\mu_i, \nu_i)$ s.t. $\mathcal{W}_d(X_i, Y_i) = \mathbb{E}d(X_i^*, Y_i^*)$. We then conclude that proof by noting that $(\{X_i^*\}_1^n, \{Y_i^*\}_1^n) \in \Pi(\mu^n, \nu^n)$. $\qquad\square$

Now we are ready to give the proof. We will do the proof for a general metric $d$. The desired result is a special case when $d$ is the Euclidean metric. We start by doing the following decomposition

$$\mathbb{E}L(\hat{\theta}_G, X) - \mathbb{E}L(\theta_0, X) = \text{I} + \text{II} + \text{III} + \text{IV} + \text{V},$$

where

$$\text{I} = \mathbb{E}L(\hat{\theta}_G, X) - \mathbb{E}\mathbb{E}_G L(\hat{\theta}_G, gX)$$

$$\text{II} = \mathbb{E}\mathbb{E}_G L(\hat{\theta}_G, gX) - \frac{1}{n}\sum_{i=1}^{n} \mathbb{E}_G L(\hat{\theta}_G, gX_i)$$

$$\text{III} = \frac{1}{n}\sum_{i=1}^{n} \mathbb{E}_G L(\hat{\theta}_G, gX_i) - \frac{1}{n}\sum_{i=1}^{n} \mathbb{E}_G L(\theta_0, gX_i)$$

$$\text{IV} = \frac{1}{n}\sum_{i=1}^{n} \mathbb{E}_G L(\theta_0, gX_i) - \mathbb{E}\mathbb{E}_G L(\theta_0, gX)$$

$$\text{V} = \mathbb{E}\mathbb{E}_G L(\theta_0, gX) - \mathbb{E}L(\theta_0, X).$$

By construction, we have III $\leq 0$ and

$$\text{II} \leq \sup_{\theta \in \Theta} \left| \frac{1}{n}\sum_{i=1}^{n} \mathbb{E}_G L(\theta, gX_i) - \mathbb{E}\mathbb{E}_G L(\theta, gX) \right|.$$

Moreover, we have

$$\text{I} + \text{V} \leq 2\sup_{\theta \in \Theta} \left| \mathbb{E}L(\theta, X) - \mathbb{E}\mathbb{E}_G L(\theta, gX) \right|,$$

which is equal to zero under exact invariance $gX =_d X$.

The term II $+$ IV is taken care of by essentially the same arguments as the proof for $\hat{\theta}_n$. One uses concentration to bound IV and uses Rademacher complexity to bound II. These arguments give

$$\text{II} + \text{IV} \leq 2\mathcal{R}_n(\bar{L} \circ \Theta) + \sqrt{\frac{2\log 2/\delta}{n}}$$

w.p. at least $1 - \delta$, where

$$\mathcal{R}_n(\bar{L} \circ \Theta) = \mathbb{E}\sup_{\theta \in \Theta} \left| \frac{1}{n}\sum_{i=1}^{n} \varepsilon_i \mathbb{E}_G L(\theta, gX_i) \right|.$$

Now we have

$$\mathcal{R}_n(\bar{L} \circ \Theta) - \mathcal{R}_n(L \circ \Theta) \leq \Delta + \mathbb{E}_g\left[ \mathbb{E}\sup_{\theta \in \Theta} \left| \frac{1}{n}\sum_{i=1}^{n} \varepsilon_i L(\theta, gX_i) \right| - \mathbb{E}\sup_{\theta \in \Theta} \left| \frac{1}{n}\sum_{i=1}^{n} \varepsilon_i L(\theta, X_i) \right| \right],$$

where we recall

$$\Delta = \mathbb{E}\sup_{\theta} |\frac{1}{n}\sum_{i=1}^{n} \varepsilon_i \mathbb{E}_g L(\theta, gX_i)| - \mathbb{E}\mathbb{E}_g \sup_{\theta \in \Theta} |\frac{1}{n}\varepsilon_i L(\theta, gX_i)| \leq 0$$

by Jensen's inequality.

By our assumption, for any $x, \tilde{x} \in \mathcal{X}, \theta \in \Theta$, we have

$$L(\theta, x) - L(\theta, \tilde{x}) \leq \|L\|_{\text{Lip}} \cdot d(x, \tilde{x})$$

for some constant $\|L\|_{\text{Lip}}$. For a fixed vector $(\varepsilon_1, ..., \varepsilon_n)$, consider the function

$$h : (x_1, ..., x_n) \mapsto \sup_{\theta \in \Theta} \left| \frac{1}{n}\sum_{i=1}^{n} \varepsilon_i L(\theta, x_i) \right|.$$

We have

$$|h(x_1, ..., x_n) - h(y_1, ..., y_n)| \leq \frac{1}{n}\sup_{\theta \in \Theta} \left| \sum_{i=1}^{n} \varepsilon_i L(\theta, x_i) - \varepsilon_i L(\theta, y_i) \right|$$

$$\leq \frac{1}{n}\|L\|_{\text{Lip}} \cdot \sum_i d(x_i, y_i).$$

That is, the function $h : \mathcal{X}^n \to \mathbb{R}$ is $(\|L\|_{\text{Lip}}/n)$-Lipschitz w.r.t. the l.s.c. metric $d_n$, defined by $d_n(\{x_i\}_1^n, \{y_i\}_1^n) = \sum_i d(x_i, y_i)$. Applying the tensorization lemma and Kantorovich-Rubinstein theorem, for arbitrary random vectors $(X_1, ..., X_n)$ and $(Y_1, ..., Y_n)$, we have

$$|\mathbb{E}h(X_1, ..., X_n) - h(Y_1, ..., Y_n)| \leq \frac{1}{n}\|L\|_{\text{Lip}} \cdot \mathcal{W}_{d^n}(\mu^n, \nu^n)$$

$$\leq \frac{1}{n}\|L\|_{\text{Lip}} \cdot \sum_{i=1}^{n} \mathcal{W}_d(X_i, Y_i).$$

Hence we arrive at

$$\mathcal{R}_n(\bar{L} \circ \Theta) - \mathcal{R}_n(L \circ \Theta) \leq \Delta + \|L\|_{\text{Lip}} \cdot \frac{1}{n}\sum_i \mathbb{E}_g \mathcal{W}_d(X_i, gX_i)$$

$$= \|L\|_{\text{Lip}} \cdot \mathbb{E}_g \mathcal{W}_d(X, gX).$$

Summarizing the above computations, we have

$$\text{II} + \text{IV} \leq 2\mathcal{R}_n(L \circ \Theta) + 2\|L\|_{\text{Lip}} \cdot \mathbb{E}_G \mathcal{W}_d(X, gX) + \sqrt{\frac{2\log 2/\delta}{n}}$$

w.p. at least $1 - \delta$.

We now bound $\text{I} + \text{V}$. We have

$$\text{I} + \text{V} \leq 2\sup_{\theta \in \Theta}\left|\mathbb{E}L(\theta, X) - \mathbb{E}\mathbb{E}_G L(\theta, gX)\right|$$

$$\leq 2\sup_{\theta \in \Theta} \mathbb{E}_G\left|\mathbb{E}L(\theta, X) - \mathbb{E}L(\theta, gX)\right|$$

$$\leq 2\|L\|_{\text{Lip}} \cdot \mathcal{W}_d(X, gX).$$

Combining the bounds for the five terms gives the desired result.

### A.3.1 The Redemacher Bound for $\hat{\theta}_n$

The results concerning $\hat{\theta}_n$ is classical. We present a proof here for completeness. We recall the classical approach of decomposing the generalization error into terms that can be bounded via concentration and Rademacher complexity [7, 60]:

$$\mathbb{E}L(\hat{\theta}_n, X) - \mathbb{E}L(\theta_0, X) = \mathbb{E}L(\hat{\theta}_n, X) - \frac{1}{n}\sum_{i=1}^{n} L(\hat{\theta}_n, X_i) + \frac{1}{n}\sum_{i=1}^{n} L(\hat{\theta}_n, X_i) - \mathbb{E}L(\theta_0, X).$$

Hence we arrive at

$$\mathbb{E}L(\hat{\theta}_n, X) - \mathbb{E}L(\theta_0, X) \leq \mathbb{E}L(\hat{\theta}_n, X) - \frac{1}{n}\sum_{i=1}^{n} L(\hat{\theta}_n, X_i) + \frac{1}{n}\sum_{i=1}^{n} L(\theta_0, X_i) - \mathbb{E}L(\theta_0, X)$$

$$\leq \sup_{\theta \in \Theta}\left|\frac{1}{n}\sum_{i=1}^{n} L(\theta, X_i) - \mathbb{E}L(\theta, X)\right| + \left(\frac{1}{n}\sum_{i=1}^{n} L(\theta_0, X_i) - \mathbb{E}L(\theta_0, X)\right),$$

where the first inequality is because $\hat{\theta}_n$ is a minimizer of the empirical risk. By McDiarmid's inequality, we have

$$\mathbb{P}(\frac{1}{n}\sum_{i=1}^{n} L(\theta_0, X_i) - \mathbb{E}L(\theta_0, X) > t) \leq \exp\{-2nt^2\}.$$

So w.p. at least $1 - \delta/2$, we have

$$\frac{1}{n}\sum_{i=1}^{n} L(\theta_0, X_i) - \mathbb{E}L(\theta_0, X) \leq \sqrt{\frac{\log 2/\delta}{2n}}.$$

It remains to control

$$\sup_{\theta\in\Theta}\left|\frac{1}{n}\sum_{i=1}^{n}L(\theta,X_i)-\mathbb{E}L(\theta,X)\right|.$$

We bound the above quantity using Rademacher complexity. The arguments are standard and can be found in many textbooks (see, e.g., [60]). Since we've assumed $L(\theta,x)\in[0,1]$, for two data sets $\{X_i\}_1^n$ and $\{\tilde{X}_i\}_1^n$ which only differ in the $i$-th coordinate, we have

$$\sup_{\theta\in\Theta}\left|\frac{1}{n}\sum_{i=1}^{n}L(\theta,X_i)-\mathbb{E}L(\theta,X)\right|-\sup_{\theta\in\Theta}\left|\frac{1}{n}\sum_{i=1}^{n}L(\theta,\tilde{X}_i)-\mathbb{E}L(\theta,X)\right|$$

$$\leq\frac{1}{n}\sup_{\theta\in\Theta}|L(\theta,X_i)-L(\theta,\tilde{X}_i)|\leq\frac{1}{n}.$$

By McDiarmid's inequality, we have

$$\mathbb{P}\left(\sup_{\theta\in\Theta}\left|\frac{1}{n}\sum_{i=1}^{n}L(\theta,X_i)-\mathbb{E}L(\theta,X)\right|-\mathbb{E}\left[\sup_{\theta\in\Theta}\left|\frac{1}{n}\sum_{i=1}^{n}L(\theta,X_i)-\mathbb{E}L(\theta,X)\right|\right]\geq t\right)\leq\exp\{-2nt^2\}.$$

It follows that w.p. $1-\delta/2$, we have

$$\sup_{\theta\in\Theta}\left|\frac{1}{n}\sum_{i=1}^{n}L(\theta,X_i)-\mathbb{E}L(\theta,X)\right|-\mathbb{E}\left[\sup_{\theta\in\Theta}\left|\frac{1}{n}\sum_{i=1}^{n}L(\theta,X_i)-\mathbb{E}L(\theta,X)\right|\right]\leq\sqrt{\frac{\log 2/\delta}{2n}}.$$

A standard symmetrization argument then shows that

$$\mathbb{E}\left[\sup_{\theta\in\Theta}\left|\frac{1}{n}\sum_{i=1}^{n}L(\theta,X_i)-\mathbb{E}L(\theta,X)\right|\right]\leq 2\mathcal{R}_n(L\circ\Theta),$$

where the Rademacher complexity of the function class $L\circ\Theta=\{x\mapsto L(\theta,x):\theta\in\Theta\}$ is defined as

$$\mathcal{R}_n(L\circ\Theta)=\mathbb{E}\sup_{\theta\in\Theta}\left|\frac{1}{n}\sum_{i=1}^{n}\varepsilon_i L(\theta,X_i)\right|,$$

where the expectation is taken over both the data and IID Rademacher random variables $\varepsilon_i$, which are independent of the data. Summarizing the above computations (along with a union bound) finishes the proof for $\hat{\theta}_n$.

### A.4 Proof of Theorem 3.5

We will do the proof with a general metric $d$. Recall that

$$\theta_G=\arg\min_{\theta\in\Theta}\mathbb{E}\mathbb{E}_g L(\theta,gX).$$

By our assumptions, we can apply Theorem 5.23 of [66] to obtain the Bahadur representation:

$$\sqrt{n}(\hat{\theta}_G-\theta_G)=\frac{1}{\sqrt{n}}V_G^{-1}\sum_{i=1}^{n}\nabla\mathbb{E}_G L(\theta_0,gX_i)+o_p(1),$$

so that we get

$$\sqrt{n}(\hat{\theta}_G-\theta_G)\Rightarrow\mathcal{N}\left(0,V_G^{-1}\mathbb{E}\left[\nabla\mathbb{E}_G L(\theta_G,gX)(\mathbb{E}_G\nabla L(\theta_G,gX))^\top\right]V_G^{-1}\right).$$

To simplify notations, we let

$$C_0=\mathrm{Cov}_X(\nabla L(\theta_0,X)),\quad C_G=\mathrm{Cov}_X(\nabla\mathbb{E}_g L(\theta_G,gX)).$$

By bias-variance decomposition, we have

$$\mathrm{MSE}_0=n^{-1}\mathrm{tr}(V_0^{-1}C_0 V_0^{-1})$$
$$\mathrm{MSE}_G=n^{-1}\mathrm{tr}(V_G^{-1}C_G V_G^{-1})+\|\theta_G-\theta_0\|^2.$$

We have

$$\operatorname{tr}(V_G^{-1} C_G V_G^{-1}) - \operatorname{tr}(V_0^{-1} C_0 V_0^{-1}) = \langle C_G, V_G^{-2} \rangle - \langle C_0, V_0^{-2} \rangle$$
$$= \langle C_G - C_0 + C_0, V_G^{-2} \rangle - \langle C_0, V_0^{-2} \rangle$$
$$= \langle C_G - C_0, V_G^{-2} \rangle + \langle C_0, V_G^{-2} - V_0^{-2} \rangle.$$

We let $M_0(X) = \nabla L(\theta_0, X) \nabla L(\theta_0, X)^\top$ and $M_G(X) = \nabla L(\theta_G, X) \nabla L(\theta_G, X)^\top$. Then we have

$$C_G - C_0 = C_G - \mathbb{E}_X M_G(X) + \mathbb{E}_X M_G(X) - C_0$$
$$= \left( C_G - \mathbb{E}_X \mathbb{E}_g M_G(gX) \right) + \mathbb{E}_g \mathbb{E}_X \left[ M_G(gX) - M_G(X) \right] + \mathbb{E}_X \left[ M_G(X) - M_0(X) \right]$$
$$= -\mathbb{E}_X \operatorname{Cov}_g(\nabla L(\theta_G, gX)) + \mathbb{E}_G \mathbb{E}_X \left[ M_G(gX) - M_G(X) \right] + \mathbb{E}_X \left[ M_G(X) - M_0(X) \right].$$

Hence we arrive at

$$n(\mathrm{MSE}_G - \mathrm{MSE}_0) = -\left\langle \mathbb{E}_X \operatorname{Cov}_G(\nabla L(\theta_G, gX)), V_G^{-2} \right\rangle + \mathrm{I} + \mathrm{II} + \mathrm{III} + \mathrm{IV},$$

where

$$\mathrm{I} = n \|\theta_G - \theta_0\|^2$$
$$\mathrm{II} = \mathbb{E}_G \mathbb{E}_X \left\langle M_G(gX) - M_G(X), V_G^{-2} \right\rangle$$
$$\mathrm{III} = \mathbb{E}_X \left\langle M_G(X) - M_0(X), V_G^{-2} \right\rangle$$
$$\mathrm{IV} = \langle C_0, V_G^{-2} - V_0^{-2} \rangle,$$

and this is the desired result.

## A.5 Proof of Theorem 3.6

We seek to control the misclassification error of the two-layer net at step $k$. By Markov's inequality, for a new sample $(X, Y)$ from the data distribution, we have

$$\mathbb{P}(Y f(x; W_k, a) \le 0) = \mathbb{P}\left( \frac{1}{1 + e^{Y f(X; W_k, a)}} \ge \frac{1}{2} \right)$$
$$\le 2\mathbb{E}\left[ -\ell'(Y f(X; W_k, a)) \right].$$

The population quantity in the RHS is decomposed by

$$\mathbb{E}\left[ -\ell'(Y f(X; W_k, a)) \right] = \mathrm{I} + \mathrm{II},$$

where

$$\mathrm{I} = \frac{1}{n} \sum_{i \in [n]} \mathbb{E}_g \left[ -\ell'(Y_i f_{i,g}(W_k)) \right]$$

and

$$\mathrm{II} = \mathbb{E}\left[ -\ell'(Y f(X; W_k, a)) \right] - \frac{1}{n} \sum_{i \in [n]} \mathbb{E}_g \left[ -\ell'(Y_i f_{i,g}(W_k)) \right]$$

The first term (optimization error) is controlled by calculations based on the Neural Tangent Kernel. The second term (generalization error) is controlled via Rademacher complexity.

We first control the first term (optimization error). In fact, everything is set up so that we can directly invoke Theorem 2.2 of [40]. We note that their result holds for any fixed dataset and there is no independence assumption. This gives the following result:

**Proposition A.2.** *Given $\varepsilon \in (0,1), \delta \in (0,1/3)$. Let*

$$\lambda = \frac{\sqrt{2\log(4n|G|/\delta)} + \log(4/\varepsilon)}{\gamma/4}, \qquad M = \frac{4096\lambda^2}{\gamma^6}.$$

*For any $m \geq M$ and any constant step size $\eta \leq 1$, w.p. $1 - 3\delta$ over the random initialization, we have*

$$\frac{1}{T}\sum_{t<T} \bar{R}_n(W_t) \leq \varepsilon, \qquad T = \lceil 2\lambda^2/(n\varepsilon)\rceil.$$

*Moreover, for any $0 \leq t < T$ and any $1 \leq s \leq m$, we have*

$$\|w_{s,t} - w_{s,0}\|_2 \leq \frac{4\lambda}{\gamma\sqrt{m}},$$

*where $w_{s,t}$ is the s-th row of the weight matrix at step $t$.*

*Proof.* This is a direct corollary of Theorem 2.2 in [40]. $\qquad\square$

Assume the above event happens. Since we've chosen $k$ to be the best iteration (with the lowest empirical loss) in the first $T$ steps. Then with the same probability as above, we have $\bar{R}_n(W_k) \leq \varepsilon$. Now, let us note that the logistic loss satisfies the following fundamental self-consistency bound: $-\ell' \leq \ell$. This shows that if the loss is small, then the magnitude of the derivative is also small. Thus on the same event, we have that the term I is also bounded,

$$\text{I} \leq \bar{R}_n(W_k) \leq \varepsilon.$$

We then control the second term (generalization error). The calculations below are similar to the proof of Theorem 4.4. We begin by decomposing

$$\text{II} = \text{II.1} + \text{II.2},$$

where

$$\text{II.1} = \mathbb{E}\left[-\ell'(Yf(X;W_k,a))\right] - \mathbb{E}\mathbb{E}_g\left[-\ell'(Yf(gX;W_k,a))\right]$$

and

$$\text{II.2} = \mathbb{E}\mathbb{E}_g\left[-\ell'(Yf(gX;W_k,a))\right] - \frac{1}{n}\sum_{i\in[n]}\mathbb{E}_g\left[-\ell'(Y_if(gX_i;W_k,a))\right].$$

We control term II.1 by exploiting the closedness between the distribution of $(X,Y)$ and that of $(gX,Y)$. Note that the Lipschitz constant of the map $x \mapsto -\ell'(yf(x;W_k,a))$ (w.r.t. the Euclidean metric on $\mathbb{R}^d$) can be computed by:

$$|\ell'(yf(x;W_k,a)) - \ell'(yf(\tilde{x};W_k,a))|$$

$$\leq \frac{1}{4}|f(x;W_k,a) - f(\tilde{x};W_k,a)|$$

$$= \frac{1}{4}\left|\frac{1}{\sqrt{m}}\sum_{s\in[m]}\sigma(w_{s,k}^\top x) - \frac{1}{\sqrt{m}}\sum_{s\in[m]}\sigma(w_{s,k}^\top \tilde{x})\right|$$

$$\leq \frac{1}{4}\frac{1}{\sqrt{m}}\sum_{s\in[m]}|w_{s,k}^\top(x-\tilde{x})|$$

$$\leq \frac{1}{4}\frac{1}{\sqrt{m}}\sum_{s\in[m]}\|w_{s,k}\|_2\|x-\tilde{x}\|_2$$

$$\leq \frac{1}{4}\frac{1}{\sqrt{m}}\sum_{s\in[m]}(\|w_{s,0}\|_2 + \|w_{s,0}-w_{s,k}\|_2)\|x-\tilde{x}\|_2$$

$$\leq \frac{1}{4}\left(\rho\sqrt{m} + \frac{1}{\sqrt{m}}\sum_{s\in[m]}\|w_{s,0}\|_2\right)\|x-\tilde{x}\|_2,$$

where $\rho = \frac{4\lambda}{\gamma\sqrt{m}}$ and the last inequality is by Proposition A.2. Note that each $\|w_{s,0}\|_2$ is 1-subgaussian as a 1-Lipschitz function of a Gaussian random vector (for example, by Theorem 2.1.12 of [64]), so that

$$\mathbb{P}\left(\frac{1}{\sqrt{m}}\sum_{s\in[m]}\|w_{s,0}\|_2 - \mathbb{E}\left[\frac{1}{\sqrt{m}}\sum_{s\in[m]}\|w_{s,0}\|_2\right] \geq t\right) \leq e^{-t^2/2}.$$

Hence w.p. at least $1 - \delta$, we have

$$\frac{1}{\sqrt{m}}\sum_{s\in[m]}\|w_{s,0}\|_2 \leq \mathbb{E}\left[\frac{1}{\sqrt{m}}\sum_{s\in[m]}\|w_{s,0}\|_2\right] + \sqrt{2\log\frac{1}{\delta}}$$

$$\leq \frac{1}{\sqrt{m}}\sum_{s\in[m]}\sqrt{\mathbb{E}\|w_{s,0}\|_2^2} + \sqrt{2\log\frac{1}{\delta}}$$

$$= \sqrt{md} + \sqrt{2\log 1/\delta}.$$

So w.p. at least $1 - \delta$, the Lipschitz constant of the map $x \mapsto -\ell'(yf(x; W_k, a))$ is bounded above by

$$\frac{1}{4}\left(\frac{4\lambda}{\gamma} + \sqrt{md} + \sqrt{2\log 1/\delta}\right).$$

Assume the above event happens (along with the previous event, the overall event happens w.p. at least $1 - 4\delta$). This information allows us to exploit the closeness between $X|Y$ and $gX|Y$. We have

$$\text{II.1} = \mathbb{E}_Y\mathbb{E}_{g\sim\mathbb{Q}}\Bigg[\mathbb{E}_{X|Y}[-\ell'(Yf(X; W_k, a))]$$

$$- \mathbb{E}_{gX|Y}[-\ell'(Yf(gX; W_k, a))]\Bigg]$$

$$\leq \mathbb{E}_Y\mathbb{E}_{g\sim\mathbb{Q}}\left[\frac{1}{4}\left(\frac{4\lambda}{\gamma} + \sqrt{md} + \sqrt{2\log 1/\delta}\right)\cdot\mathcal{W}_1(X|Y, gX|Y)\right]$$

$$= \frac{1}{4}\left(\frac{4\lambda}{\gamma} + \sqrt{md} + \sqrt{2\log 1/\delta}\right)\cdot\mathbb{E}_Y\mathbb{E}_{g\sim\mathbb{Q}}\mathcal{W}_1(X|Y, gX|Y),$$

where we let $X|Y$ to denote the conditional distribution of $X$ given $Y$, and the inequality is by the dual representation of the Wasserstein distance. Note that under exact invariance, II.1 = 0.

The term II.2 is controlled by standard results on Rademacher complexity. Indeed, by the same arguments as in the proof of Theorem 6.4 of the main manuscript, w.p. at least $1 - \delta$, we have

$$\text{II.2} \leq 2\bar{\mathcal{R}}_n + \sqrt{\frac{\log 2/\delta}{2n}}.$$

Taking a union bound (now w.p. at least $1 - 5\delta$), we have proved the generalization error bound.

Finally, we prove the bound on $\bar{\mathcal{R}}_n - \mathcal{R}_n$. Under exact invariance, Jensen's inequality gives $\bar{\mathcal{R}}_n \leq \mathcal{R}_n$. However, under approximate invariance, we have an extra bias term. We have

$$\bar{\mathcal{R}}_n - \mathcal{R}_n = \Delta + \mathbb{E}\mathbb{E}_g \sup_{W\in\mathscr{W}_\rho}\left|\frac{1}{n}\varepsilon_i\left[-\ell'(Y_if_{i,g}(W))\right]\right| - \mathcal{R}_n.$$

where

$$\Delta = \mathbb{E}\sup_{W\in\mathscr{W}_\rho}\left|\frac{1}{n}\varepsilon_i\mathbb{E}_g\left[-\ell'(Y_if_{i,g}(W))\right]\right| - \mathbb{E}\mathbb{E}_g\sup_{W\in\mathscr{W}_\rho}\left|\frac{1}{n}\varepsilon_i\left[-\ell'(Y_if_{i,g}(W))\right]\right| \leq 0$$

by Jensen's inequality. Now by the computations when bounding term II.1 and the arguments in the proof of Theorem 4.4, we have

$$\mathbb{E}\mathbb{E}_g\sup_{W\in\mathscr{W}_\rho}\left|\frac{1}{n}\varepsilon_i\left[-\ell'(Y_if_{i,g}(W))\right]\right| - \mathcal{R}_n \leq \frac{1}{4}\left(\frac{4\lambda}{\gamma} + \sqrt{md} + \sqrt{2\log 1/\delta}\right)$$

$$\cdot\mathbb{E}_Y\mathbb{E}_{g\sim\mathbb{Q}}\mathcal{W}_1(X|Y, gX|Y)$$

w.p. at least $1 - \delta$. Combining the above bounds finishes the proof.

## A.6 Proof of Theorem 4.1

We first present two useful lemmas. The first one can is essentially Lemma 3.2 under exact invariance:

**Lemma A.3** (Invariance lemma). *In the setting of Lemma 3.2, assume exact invariance holds. Then*

1. *For any $x$, $\bar{f}(x)$ is the conditional expectation of $f(X)$, conditional on the orbit: $\bar{f}(x) = \mathbb{E}[f(X)|X \in Gx]$, where $Gx := \{gx : g \in G\}$;*
2. *Therefore, by the law of total expectation, the mean of $\bar{f}(X)$ and $f(X)$ coincide: $\mathbb{E}_{X \sim \mathbb{P}} f(X) = \mathbb{E}_{X \sim \mathbb{P}} \bar{f}(X)$;*
3. *By the law of total covariance, the covariance of $f(X)$ can be decomposed as $\mathrm{Cov}_{X \sim \mathbb{P}} f(X) = \mathrm{Cov}_{X \sim \mathbb{P}} \bar{f}(X) + \mathbb{E}_{X \sim \mathbb{P}} \mathrm{Cov}_{g \sim \mathbb{Q}} f(gX)$;*
4. *Let $\varphi$ be any real-valued convex function. Then $\mathbb{E}_{X \sim \mathbb{P}}[\varphi(f(X))] \geq \mathbb{E}_{X \sim \mathbb{P}}[\varphi(\bar{f}(X))]$.*

*Proof.* We first prove part 1. Let $x$ be fixed. Let $A = \{X \in Gx\}$. It suffices to show

$$\int_A \mathbb{E}_g f(gx) d\mathbb{P}(X) = \int_A f(X) d\mathbb{P}(X).$$

For an arbitrary $g \in G$, the RHS above is equal to

$$\int_A f(X) d\mathbb{P}(X) = \int f(gX) \mathbb{1}\{gX \in Gx\} d\mathbb{P}(X) = \int f(gX) \mathbb{1}\{X \in Gx\} d\mathbb{P}(X),$$

where the first equality is by the exact invariance, and the second equality is by the definition of the orbit. Taking expectation w.r.t. $\mathbb{Q}$, we get

$$\int_A f(X) d\mathbb{P}(X) = \int_G \int f(gX) \mathbb{1}\{X \in Gx\} d\mathbb{P}(X) d\mathbb{Q}(g).$$

On the event $A$, there exists $g_X^*$, potentially depending on $X$, s.t. $X = g_X^* x$. Hence, we have

$$
\begin{aligned}
\int_A f(X) d\mathbb{P}(X) &= \int_G \int f(g \circ g_X^* x) \mathbb{1}\{X \in Gx\} d\mathbb{P}(X) d\mathbb{Q}(g) \\
&= \int \int_G f(g \circ g_X^* x) d\mathbb{Q}(g) \mathbb{1}\{X \in Gx\} d\mathbb{P}(X) \\
&= \int \int_G f(gx) d\mathbb{Q}(g) \mathbb{1}\{X \in Gx\} d\mathbb{P}(X) \\
&= \int_A \mathbb{E}_g f(gx) d\mathbb{P}(X),
\end{aligned}
$$

where the second equality is by Fubini's theorem, and the third inequality is due to the translation invariant property of the Haar measure.

Part 2 follows by law of total expectation along with the above point.

Part 3 follows directly from part 1 and the law of the total covariance applied to the random variable $f(gX)$, where $g \sim \mathbb{Q}, X \sim \mathbb{P}$.

Part 4 follows from Jensen's inequality. $\qquad\square$

The next lemma says that if Assumptions A and B hold for the pair $(\theta_0, L)$, then they also hold for the pair $(\theta_G, \bar{L})$ under exact invariance:

**Lemma A.4** (Regularity of the augmented loss). *For each $\theta$, assume the map $(X, g) \mapsto L(\theta, gX)$ is in $L^1(\mathbb{P} \times \mathbb{Q})$. Assume exact invariance holds. If the pair $(\theta_0, L)$ satisfies Assumption A and B, then the two assumptions also hold for the pair $(\theta_G, \bar{L})$.*

*Proof.* By exact invariance and Fubuni's theorem, it is clear that $\mathbb{E}\bar{L}(\theta, X) = \mathbb{E}L(\theta, X)$ for any $\theta \in \Theta$. Hence $\theta_G = \theta_0$ and Assumption $A$ is verified for $(\theta_G, \bar{L})$. We now verify the five parts of Assumption B.

For part 1, we have

$$\sup_{\theta \in \Theta} \left| \frac{1}{n} \sum_{i=1}^{n} \mathbb{E}\bar{L}(\theta, X_i) - \mathbb{E}\bar{L}(\theta, X) \right| = \sup_{\theta \in \Theta} \left| \mathbb{E}_g[\frac{1}{n} \sum_{i=1}^{n} L(\theta, gX_i) - \mathbb{E}L(\theta, gX)] \right|$$

$$\leq \sup_{\theta \in \Theta} \mathbb{E}_g \left| \frac{1}{n} \sum_{i=1}^{n} L(\theta, gX_i) - \mathbb{E}L(\theta, gX) \right|$$

$$\leq \mathbb{E}_g \sup_{\theta \in \Theta} \left| \frac{1}{n} \sum_{i=1}^{n} L(\theta, gX_i) - \mathbb{E}L(\theta, gX) \right|$$

$$= o_p(1),$$

where the two inequalities is by Jensen's inequality, and the convergence statement is true because of the exact invariance and the fact that the original loss satisfies part 1 of Assumption B.

Part 2 is true because we have assumed that the action $x \mapsto gx$ is continuous.

For part 3, since $(\theta_0, L)$ satisfies this assumption, we know that on an event with full probability, we have

$$\lim_{\delta \to 0} \frac{\left| L(\theta_0 + \delta, gX) - L(\theta_0, gX) - \delta^\top \nabla L(\theta_0, gX) \right|}{\|\delta\|} = 0.$$

Now we have

$$\frac{\left| \mathbb{E}_g L(\theta_0 + \delta, gX) - \mathbb{E}_g L(\theta_0, gX) - \delta^\top \mathbb{E}_g \nabla L(\theta_0, gX) \right|}{\|\delta\|}$$

$$\leq \frac{\mathbb{E}_g \left| L(\theta_0 + \delta, gX) - L(\theta_0, gX) - \delta^\top \nabla L(\theta_0, gX) \right|}{\|\delta\|}$$

$$\leq \mathbb{E}_g[\dot{L}(gX) + \|\nabla L(\theta_0, gX)\|],$$

where the first inequality is by Jensen's inequality, and the second inequality is by part 4 applied to $(\theta_0, L)$. We have assumed that $\mathbb{E}[\dot{L}(X)^2] < \infty$, and hence so does $\mathbb{E}[\dot{L}(X)]$. By exact invariance, we have

$$\mathbb{E}_X \mathbb{E}_g[\dot{L}(gX)] = \mathbb{E}_g \mathbb{E}_X[\dot{L}(gX)] = \mathbb{E}[\dot{L}(X)] < \infty,$$

which implies $\mathbb{E}_g[\dot{L}(gx)] < \infty$ for $\mathbb{P}$-a.e. $x$. On the other hand, part 5 applied to $(\theta_0, L)$ implies the existence of $\mathbb{E}\nabla L(\theta_0, X)\nabla L(\theta_0, X)^\top$, and hence $\mathbb{E}\|\nabla L(\theta_0, X)\|^2 < \infty$ and so does $\mathbb{E}\|\nabla L(\theta_0, X)\|$. Now a similar argument shows that under exact invariance, we have

$$\mathbb{E}_g \|\nabla L(\theta_0, gx)\| < \infty.$$

for $\mathbb{P}$-a.e. $x$. Hence we can apply dominated convergence theorem to conclude that

$$\lim_{\delta \to 0} \frac{\left| \mathbb{E}_g L(\theta_0 + \delta, gX) - \mathbb{E}_g L(\theta_0, gX) - \delta^\top \mathbb{E}_g \nabla L(\theta_0, gX) \right|}{\|\delta\|}$$

$$\leq \mathbb{E}_g \lim_{\delta \to 0} \frac{\left| L(\theta_0 + \delta, gX) - L(\theta_0, gX) - \delta^\top \nabla L(\theta_0, gX) \right|}{\|\delta\|}$$

$$= 0,$$

which implies that $\theta \mapsto \bar{L}(\theta, x)$ is indeed differentiable at $\theta_0$.

For part 4, by assumption, we have

$$|L(\theta_1, gx) - L(\theta_2, gx)| \leq \dot{L}(gx)\|\theta_1 - \theta_2\|$$

for almost every $x$ and every $\theta_1, \theta_2$ in a neighborhood of $\theta_0 = \theta_G$. Taking expectation w.r.t. $g \sim \mathbb{Q}$, we get

$$|\bar{L}(\theta_1, x) - \bar{L}(\theta_2, x)| \le \mathbb{E}_g |L(\theta_1, gx) - L(\theta_2, gx)|$$
$$\le \mathbb{E}_g \dot{L}(gx) \|\theta_1 - \theta_2\|.$$

Now it suffices to show $x \mapsto \mathbb{E}_g \dot{L}(gx)$ is in $L^2(\mathbb{P})$. This is true by an application of Jensen's inequality and exact invariance:

$$\mathbb{E}_X[(\mathbb{E}_g \dot{L}(gX))^2] \le \mathbb{E}_X \mathbb{E}_g[(\dot{L}(gX))^2]$$
$$= \mathbb{E}_g \mathbb{E}_X[(\dot{L}(gX))^2]$$
$$= \mathbb{E}_g \mathbb{E} \dot{L}^2$$
$$= \mathbb{E} \dot{L}^2$$
$$< \infty.$$

Part 5 is true because under exact invariance, we have $\mathbb{E}L(\theta, X) = \mathbb{E}\bar{L}(\theta, X)$. $\qquad\square$

We now present the proof of Theorem 4.1.

*Proof of Theorem 4.1.* The results concerning $\hat{\theta}_n$ is classical (see, e.g., Theorem 5.23 of [66]). By Lemma 3.4, we can apply Theorem 5.23 of [66] to the pair $(\theta_G = \theta_0, \bar{L})$ to conclude that

$$\sqrt{n}(\hat{\theta}_{nG} - \theta_0) = \frac{1}{\sqrt{n}} V_{\theta_0}^{-1} \sum_{i=1}^{n} \nabla \bar{L}(\theta_0, X_i) + o_p(1).$$

Hence the asymptotic normality of $\hat{\theta}_{n,G}$ follows and we have

$$\Sigma_G = V_{\theta_0}^{-1} \mathbb{E}[\bar{L}(\theta_0, X) \bar{L}(\theta_0, X)^\top] V_{\theta_0}^{-1}.$$

The final representation of $\Sigma_G$ follows from part 3 of Lemma A.3. $\qquad\square$

## A.7 Proof of Theorem 4.2

For notational simplicity, we will use $W$ and $\theta$ interchangeably when there is no ambiguity. Recall the regression function is $f(W, X) = \mathbf{1}^\top \sigma(WX)$. Let $\theta_0$ be the ground truth weight matrix. The population risk is

$$\mathbb{E}L(\theta, X, Y) = \mathbb{E}(Y - f(\theta_0, X) + f(\theta_0, X) - f(\theta, X))^2$$
$$= \mathbb{E}(f(\theta_0, X) - f(\theta, X))^2 + \gamma^2.$$

Under current assumptions, the minimizer $\hat{W}_n$ of $W \mapsto n^{-1} \sum_{i=1}^{n} L(W, X_i, Y_i)$ is consistent (see, e.g., Example 5.27 of [66]). Meanwhile, under current assumptions, we have the following second-order expansion:

$$\mathbb{E}L(\theta, X, Y) = \mathbb{E}L(\theta_0, X, Y) + \frac{1}{2} \mathbb{E}\left[ (\theta - \theta_0)^\top \left( 2 \nabla f(\theta_0, X) \nabla f(\theta_0, X)^\top \right) (\theta - \theta_0) \right]$$
$$+ o(\|\theta - \theta_0\|^2),$$

where $\mathbb{E}L(\theta_0, X, Y) = \gamma^2$ and $\nabla f(\theta, X)$ is the gradient w.r.t. $\theta$. This suggests that we can apply Theorem 3.5 with $V_{\theta_0} = 2\mathbb{E}\nabla f(\theta_0, X) \nabla f(\theta_0, X)^\top$ and $\nabla L(\theta_0, X, Y) = -2(Y - f(\theta_0, X)) \nabla f(\theta_0, X) = -2\varepsilon \nabla f(\theta_0, X)$, which gives (with the Fisher information $I_\theta = \mathbb{E}\nabla f(\theta, X) \nabla f(\theta, X)^\top$)

$$\sqrt{n}(\hat{\theta}_{\text{ERM}} - \theta_0) \Rightarrow \mathcal{N}\left( 0, V_{\theta_0}^{-1} \mathbb{E}\left[ 4\varepsilon^2 \nabla f(\theta_0, X) \nabla f(\theta_0, X)^\top \right] V_{\theta_0}^{-1} \right)$$
$$=_d \mathcal{N}\left( 0, \sigma^2 I_{\theta_0}^{-1} \right).$$

On the other hand, the augmented ERM estimator is the minimizer $W_{n,G}$ of $W \mapsto \sum_{i=1}^{n} \mathbb{E}_g L(W, gX_i, Y_i)$. Now applying Theorem 3.5 gives

$$\sqrt{n}(W_{n,G} - \theta_0) \Rightarrow \mathcal{N}(0, \Sigma_G),$$

with the asymptotic covariance being

$$\Sigma_G = \gamma^2 I_{\theta_0}^{-1} - V_{\theta_0}^{-1} \mathbb{E}\left[\mathrm{Cov}_g \nabla L(\theta_0, gX, Y)\right] V_{\theta_0}^{-1}$$

$$= \gamma^2 I_{\theta_0}^{-1} - I_{\theta_0}^{-1} \mathbb{E}\left[\mathrm{Cov}_g(Y - f(\theta_0, gX)) \nabla f(\theta_0, gX)\right] I_{\theta_0}^{-1}$$

$$= \gamma^2 I_{\theta_0}^{-1} - I_{\theta_0}^{-1} \mathbb{E}\left[\varepsilon^2 \mathrm{Cov}_g \nabla f(\theta_0, gX)\right] I_{\theta_0}^{-1}$$

$$= \gamma^2 \cdot \left(I_{\theta_0}^{-1} - I_{\theta_0}^{-1} \mathbb{E}\left[\mathrm{Cov}_g \nabla f(\theta_0, gX)\right] I_{\theta_0}^{-1}\right),$$

where we used $f(\theta_0, gx) = f(\theta_0, x)$ (which is due to exact invariance) in the second to last line. Using Lemma 3.1, we can also write

$$\Sigma_G$$

$$= \gamma^2 I_{\theta_0}^{-1} \left(I_{\theta_0} - \mathbb{E}\left[\mathrm{Cov}_g \nabla f(\theta_0, gX)\right]\right) I_{\theta_0}^{-1}$$

$$= \gamma^2 I_{\theta_0}^{-1} \left(\mathrm{Cov}_X \nabla f(\theta_0, X) - \mathbb{E}\left[\mathrm{Cov}_g \nabla f(\theta_0, gX)\right]\right) I_{\theta_0}^{-1}$$

$$= \gamma^2 I_{\theta_0}^{-1} \bar{I}_{\theta_0} I_{\theta_0}^{-1},$$

where $\bar{I}_{\theta_0}$ is the "averaged Fisher information", defined as

$$\bar{I}_{\theta_0} = \mathrm{Cov}_X[\mathbb{E}_g \nabla f(\theta_0, gX)]$$

$$= \mathbb{E}_X\left[\left(\mathbb{E}_g \nabla f(\theta_0, gX)\right)\left(\mathbb{E}_g \nabla f(\theta_0, gX)\right)^{\top}\right].$$

This finishes the proof for the asymptotic normality result.

We now prove the expressions for $I_W$ and $\bar{I}_W$. We have

$$\nabla f(W, x) = \sigma'(Wx) \cdot x^{\top} \in \mathbb{R}^{m \times d}.$$

We can think of the Fisher information matrix $I_\theta = \mathbb{E}\nabla f(\theta, X)\nabla f(\theta, X)^{\top}$ as a $d \times d \times d \times d$ tensor, i.e,

$$I_W = \mathbb{E}(\sigma'(WX) \cdot X^{\top}) \otimes (\sigma'(WX) \cdot X^{\top})$$

$$= \mathbb{E}(\sigma'(WX) \otimes \sigma'(WX)) \cdot (X \otimes X)^{\top}.$$

The $(i, j, i', j')$-th entries of this tensor is

$$I_W(i, j, i', j') = \mathbb{E}\sigma'(W_i^{\top} X)\sigma'(W_{i'}^{\top} X) \cdot X_j X_{j'}.$$

For a quadratic activation function, $\sigma(x) = x^2/2$, we have

$$I_W = \mathbb{E}(WXX^{\top}) \otimes (WXX^{\top})$$

$$= (W \otimes W) \cdot \mathbb{E}(XX^{\top} \otimes XX^{\top}).$$

The group acts by $gx = T_g x$, where $T_g$ is an operator that shifts a vector circularly by $g$ units. We can then write the neural network $f(W, x) = \sum_{i=1}^{p} h(W_i; x)$ as a sum, where $h(a; x) = \sigma(a^{\top}x)$. Therefore, the invariant function corresponding to $f_W$ can also be written in terms of the corresponding invariant functions corresponding to the $h$-s:

$$\bar{f}(W, x) = \frac{1}{d}\sum_{g=1}^{d} f(W, T_g x) = \sum_{i=1}^{p} \bar{h}(W_i; x).$$

where $\bar{h}(a;x) = \frac{1}{d}\sum_{g=1}^d h(a;T_g x)$. We can use this representation to calculate the gradient. We first notice $\nabla h(a;x) = \sigma'(a^\top x)x$. Thus,

$$\nabla \bar{h}(a;x) = \frac{1}{d}\sum_{g=1}^d \nabla h(a;T_g x)$$

$$= \frac{1}{d}\sum_{g=1}^d \sigma'(a^\top T_g x)T_g x$$

$$= \frac{1}{d}C_x \cdot \sigma'(C_x^\top a).$$

Here $C_x$ is the circulant matrix

$$C_x = [x, T_1 x, \ldots, T_{d-1} x] = \begin{bmatrix} x_1, & x_d, & \ldots, & x_{d-1} \\ x_2, & x_1, & \ldots, & x_d \\ & & \ldots & \\ x_d, & x_{d-1}, & \ldots, & x_1 \end{bmatrix}.$$

Hence the gradient of the invariant neural network $\bar{f}(W,x)$ can be written as a matrix-vector product

$$\nabla \bar{f}(W,x) = \begin{bmatrix} \nabla \bar{h}(W_1;x)^\top \\ \ldots \\ \nabla \bar{h}(W_p;x)^\top \end{bmatrix}$$

$$= \frac{1}{d}\begin{bmatrix} \sigma'(W_1^\top C_x)\cdot C_x^\top \\ \ldots \\ \sigma'(W_p^\top C_x)\cdot C_x^\top \end{bmatrix}$$

$$= \frac{1}{d}\sigma'(WC_x)\cdot C_x^\top.$$

So the Fisher information can also expressed in terms of matrix products

$$\bar{I}_W = \mathbb{E}(\sigma'(WC_X)\cdot C_X^\top)\otimes(\sigma'(WC_X)\cdot C_X^\top)$$

$$= \mathbb{E}(\sigma'(WC_X)\otimes \sigma'(WC_X))\cdot(C_X\otimes C_X)^\top.$$

For quadratic activation functions, we have

$$\bar{I}_W = \frac{1}{d^2}\mathbb{E}(WC_X C_X^\top)\otimes(WC_X C_X^\top)$$

$$= (W\otimes W)\cdot\frac{1}{d^2}\mathbb{E}(C_X C_X^\top\otimes C_X C_X^\top)$$

$$= (W\otimes W)\cdot\frac{1}{d^2}\mathbb{E}(C_X\otimes C_X)\cdot(C_X\otimes C_X)^\top.$$

This finishes the proof for the expression of Fisher information matrices.

Finally, we prove expressions for the expected Fisher information matrices under the assumption of $X \sim \mathcal{N}(0, I_d)$. The $(i,j,i',j')$-th entry of $\mathbb{E}[XX^\top\otimes XX^\top]$ is given by

$$X_i X_j X_{i'} X_{j'},$$

whose expectation is easy to compute by the fact that $\mathbb{E}X_i = 0, \mathbb{E}X_i^2 = 1, \mathbb{E}X_i^3 = 0$ and $\mathbb{E}X_i^4 = 3$.

On the other hand, we can express $\bar{I}_W$ in a simpler form using the discrete Fourier transform. Let $F$ be the $d\times d$ Discrete Fourier Transform (DFT) matrix, with entries $F_{j,k} = d^{-1/2}\exp(-2\pi i/d\cdot(j-1)(k-1))$. Then $Fx$ is called the DFT of the vector $x$, and $F^{-1}y = F^*y$ is called the inverse DFT. The DFT matrix is a unitary matrix with $FF^* = F^*F = I_d$. Thus $F^{-1} = F^*$. It is also a symmetric matrix with $F^\top = F$. Then the circular matrix can be diagonalized as

$$\frac{1}{\sqrt{d}}C_x = F^*\operatorname{diag}(Fx)F.$$

The eigenvalues of $d^{-1/2}C_x$ are the entries of $Fx$, with eigenvectors the corresponding columns of $F$. So we can write, with $D := \operatorname{diag}(FX)$,

$$
\begin{aligned}
d^{-1}C_X \otimes C_X = F^*DF \otimes F^*DF \\
= (F \otimes F)^* \cdot (D \otimes D) \cdot (F \otimes F) \\
= F_2^* D_2 F_2,
\end{aligned}
$$

where $F_2 = F \otimes F$, and $D_2 = D \otimes D$ is a diagonal matrix. So

$$
\begin{aligned}
\frac{\mathbb{E}(C_X \otimes C_X) \cdot (C_X \otimes C_X)^\top}{d^2} = \mathbb{E}F_2^* D_2 F_2 \cdot (F_2^* D_2 F_2)^\top \\
= \mathbb{E}F_2^* D_2 F_2 \cdot F_2^\top D_2 F_2^{*,T} \\
= F_2^* \cdot \mathbb{E}D_2 F_2^2 D_2 \cdot F_2^*.
\end{aligned}
$$

Here we used that $F = F^\top$, hence $F_2^\top = (F \otimes F)^\top = F^\top \otimes F^\top = F_2$. Now, $D_2$ can be viewed as a $d^2 \times d^2$ matrix, with diagonal entries $D_2(i, j, i, j) = D_i D_j = F_i^\top X \cdot F_j^\top X$, where $F_i$ are the rows (which are also equal to the columns) of the DFT. Thus the inner expectation can be written as an elementwise product (also known as Hadamard or odot product)

$$
\mathbb{E}D_2 F_2^2 D_2 = F_2^2 \odot \mathbb{E}D_2 D_2^\top.
$$

So we only need to calculate the 4th order moment tensor $M$ of the Fourier transform $FX$,

$$
M_{iji'j'} = \mathbb{E}F_i^\top X \cdot F_j^\top X \cdot F_{i'}^\top X \cdot F_{j'}^\top X.
$$

Let us write $r := FX$. Then by Wick's formula,

$$
\mathbb{E}f_i f_j f_{i'} f_{j'} = \mathbb{E}f_i f_j \cdot \mathbb{E}f_{i'} f_{j'} + \mathbb{E}f_i f_{j'} \cdot \mathbb{E}f_{i'} f_j + \mathbb{E}f_i f_{i'} \cdot \mathbb{E}f_i f_{j'}.
$$

Now

$$
\mathbb{E}f_i f_j = \mathbb{E}F_i^\top X \cdot F_j^\top X = F_i^\top \cdot \mathbb{E}XX^\top \cdot F_j = F_i^\top F_j.
$$

Hence

$$
M_{iji'j'} = F_i^\top F_j \cdot F_{i'}^\top F_{j'} + F_i^\top F_{j'} \cdot F_{i'}^\top F_j + F_i^\top F_{i'} \cdot F_i^\top F_{j'}.
$$

This leads to a completely explicit expression for the average information. Recall $F_2 = F \otimes F$, and $M$ is the $d^2 \times d^2$ tensor with entries given above. Then

$$
\bar{I}_W = (W \otimes W) \cdot F_2^* \cdot (F_2^2 \odot M) \cdot F_2^*,
$$

completing the proof of this theorem.

# B    Data Augmentation with General Estimators: Linear Regression

In this section, we present a tight analysis of the augmented estimator $\hat{\theta}_G(X) = \mathbb{E}_{g \sim \mathbb{Q}}\hat{\theta}(gX)$ introduced in Proposition 3.3, under the linear regression setup.

We consider the classical linear regression model

$$
Y = X^\top \beta + \varepsilon, \qquad \beta \in \mathbb{R}^p.
$$

Let $\gamma^2 = \mathbb{E}\varepsilon^2$. We will assume that the action is linear, so that $g$ can be represented as a $p \times p$ matrix. If we augment by a single fixed $g$, we get

$$
\arg\min \|y - Xg^\top \beta\|_2^2 = ((Xg^\top)^\top Xg^\top)^{-1}(Xg^\top)^\top y.
$$

Following the ideas on augmentation distribution, we can then average the above estimator over $g \sim \mathbb{Q}$ to obtain

$$
\hat{\beta}_{\text{aDIST}} = \mathbb{E}_{g \sim \mathbb{Q}}\left[((Xg^\top)^\top Xg^\top)^{-1}(Xg^\top)^\top y\right].
$$

On the other hand, let us consider the estimator arising from constrained ERM. Under exact invariance, we have

$$
x^\top \beta = (gx)^\top \beta
$$

for $\mathbb{P}_X$-a.e. $x$ and $\mathbb{Q}$-a.e. $g$. This is a set of *linear constraints* on the regression coefficient $\beta$. Formally, supposing that $x$ can take any value (i.e., $\mathbb{P}_X$ has mass on the entire $\mathbb{R}^p$), we conclude that $\beta$ is constrained to be in the invariant parameter subspace $\Theta_G$, which is a linear subspace defined by

$$\Theta_G = \{v : g^\top v = v, \forall g \in G\}.$$

If $x$ can only take values in a smaller subset of $\mathbb{R}^p$, then we get fewer constraints. So the constrained ERM, defined as

$$\hat{\beta}_{\text{cERM}} = \arg\min_\beta \|y - X\beta\|_2^2 \ \ s.t. \ (g^\top - I_p)\beta = 0 \ \forall g \in G,$$

can in principle, be solved via convex optimization.

Intuitively, we expect both $\hat{\beta}_{\text{aDIST}}$ and $\hat{\beta}_{\text{cERM}}$ to be better than the vanilla ERM

$$\hat{\beta}_{\text{ERM}} = \arg\min_\beta \|y - X\beta\|_2^2,$$

Let $r_{\text{ERM}}, r_{\text{aDIST}}, r_{\text{cERM}}$ be the mean squared errors of the three estimators. We summarize the relationship between the three estimators in the following proposition:

**Proposition B.1** (Comparison between ERM, aDIST and cERM in linear regression). *Let the action of $G$ be linear. Then:*

1. *Denote $v_j \in \mathbb{R}^p$ as the $j$-th eigenvector of $X^T X$ and $d_j^2$ as the corresponding eigenvalue. We have*

$$r_{\text{ERM}} = \gamma^2 \operatorname{tr}[X^\top X]^{-1} = \gamma^2 \sum_{j=1}^p d_j^{-2}, \qquad r_{\text{aDIST}} = \gamma^2 \sum_{j=1}^p d_j^{-2} \|\mathcal{G}^\top v_j\|_2^2,$$

   *where $\mathcal{G} = \mathbb{E}_{g \sim \mathbb{Q}}[g]$.*

2. *If $G$ acts orthogonally, then $r_{\text{aDIST}} \leq r_{\text{ERM}}$.*

3. *If $G$ is the permutation group over $\{1, ..., p\}$, then*

$$r_{\text{aDIST}} = \gamma^2 p^{-1} 1_p^\top (X^\top X)^{-1} 1_p, \qquad r_{\text{cERM}} = \gamma^2 p (1_p^\top X^\top X 1_p)^{-1}.$$

   *Furthermore, if $X$ is an orthogonal design so that $X^\top X = I_p$, we have*

$$r_{\text{ERM}} = p\gamma^2, \qquad r_{\text{aDIST}} = r_{\text{cERM}} = \gamma^2.$$

*Proof.* For part 1, we have

$$\begin{aligned}
\hat{\beta}_{\text{aDIST}} &= \mathbb{E}_g \left[ ((Xg^\top)^\top Xg^\top)^{-1} (Xg^\top)^\top y \right] \\
&= \mathbb{E}_g \left[ (gX^\top Xg^\top)^{-1} gX^\top (Xg^\top \beta + \varepsilon) \right] \\
&= \beta + \mathbb{E}_g \left[ (gX^\top Xg^\top)^{-1} gX^\top \varepsilon \right] \\
&= \beta + \mathbb{E}_g \left[ g^{-1} (X^\top X)^{-1} g^{-1} gX^\top \varepsilon \right] \\
&= \beta + \mathbb{E}_g[g^{-1}] (X^\top X)^{-1} X^\top \varepsilon \\
&= \beta + \mathcal{G}^\top (X^\top X)^{-1} X^\top \varepsilon.
\end{aligned}$$

Let $X = UDV^\top$ be a SVD of $X$, where $V \in \mathbb{R}^{p \times p}$ is unitary. Note that $\hat{\beta}_{\text{aDIST}}$ is unbiased, so its $\ell_2$ risk is

$$\begin{aligned}
r_{\text{aDIST}} &= \gamma^2 \operatorname{tr}(\operatorname{Var}(\hat{\beta}_{\text{aDIST}})) = \gamma^2 \operatorname{tr}(\mathcal{G}^\top (X^\top X)^{-1} \mathcal{G}) \\
&= \gamma^2 \operatorname{tr}(\mathcal{G}^\top V D^{-2} V^\top \mathcal{G}) = \gamma^2 \operatorname{tr}(D^{-2} V^\top \mathcal{G} \mathcal{G}^\top V) \\
&= \gamma^2 \sum_{j=1}^p d_j^{-2} e_j^\top V^\top \mathcal{G} \mathcal{G}^\top V e_j = \gamma^2 \sum_{j=1}^p d_j^{-2} \|\mathcal{G}^\top v_j\|_2^2,
\end{aligned}$$

where $v_j \in \mathbb{R}^p$ is $j$-th eigenvector of $X^\top X$ and $d_j^2$ is $j$-th eigenvalue of $X^\top X$. As a comparison, for the usual ERM, we have

$$\hat{\beta}_{\text{ERM}} = (X^\top X)^{-1} X^\top y = \beta + (X^\top X)^{-1} X^\top \varepsilon,$$

so its $\ell_2$ risk is

$$r_{\text{ERM}} = \gamma^2 \operatorname{tr}((X^\top X)^{-1}) = \gamma^2 \sum_{j=1}^{p} d_j^{-2}.$$

So part 1 is proved.

We now prove part 2. For $r_{\text{aDIST}} \leq r_{\text{ERM}}$ we need to show

$$\operatorname{tr}((X^\top X)^{-1} \mathcal{G}\mathcal{G}^\top) \leq \operatorname{tr}((X^\top X)^{-1}).$$

A sufficient condition is that, in the partial ordering of positive semidefinite matrices,

$$\mathcal{G}\mathcal{G}^\top \leq I_p.$$

This is equivalent to the claim that for all $v$ $\|\mathbb{E}_g g^\top v\|^2 \leq \|v\|^2$. However, by Jensen's inequality, $\|\mathbb{E}_g g^\top v\|^2 \leq \mathbb{E}_G \|g^\top v\|^2$. Since $G$ is a subgroup of the orthogonal group, we have $\|g^\top v\|^2 = \|v\|^2$, which finishes the proof for part 2.

We finally prove part 3. We assume $G$ is the permutation group. This group is clearly a subgroup of the orthogonal group. Note that invariance w.r.t. $G$ implies that the true parameter is a multiple of the all ones vector: $\beta = 1_p b$. So we have

$$\hat{\beta}_{\text{cERM}} = 1_p \hat{b}, \qquad \hat{b} = \arg\min \|y - X 1_p b\|_2^2.$$

Solving the least-squares equation gives

$$\hat{b} = \frac{1^\top X^\top y}{1_p^\top X^\top X 1_p}.$$

The risk of estimating $b$ is then $\gamma^2 (1_p^\top X^\top X 1_p)^{-1}$, so that the risk of estimating $\beta$ by $1_p \hat{b}$ is

$$r_{\text{cERM}} = \gamma^2 p (1_p^\top X^\top X 1_p)^{-1}.$$

Finally, we have

$$r_{\text{aDIST}} = \frac{\gamma^2}{p^2} \operatorname{tr}(1_p 1_p^\top (X^\top X)^{-1} 1_p 1_p^\top) = \frac{\gamma^2}{p} 1_p^\top (X^\top X)^{-1} 1_p,$$

which is equal to $r_{\text{cERM}}$ if $X^\top X = I_p$. $\qquad\qquad\square$

In general, constrained ERM can be even more efficient than the estimator obtained by the augmentation distribution. However, by the third point in the above proposition, in the special case where $G$ is the permutation group, we have $r_{\text{aDIST}} = r_{\text{cERM}} \ll r_{\text{ERM}}$ when the dimension $p$ is large. A direct extension of the above proposition shows that such a phenomenon occurs when $G$ is the permutation group on a subset of $\{1, \ldots, p\}$. There are several other subgroups of interest of the permutation group, including the group of cyclic permutations and the group that contains the identity and the operation that "flips" or reverses each vector.

We note briefly that the above results apply *mutatis mutandis* to logistic regression. There, the outcome $Y \in \{-1, 1\}$ is binary, and $P(Y = 1 | X = x) = \sigma(x^\top \beta)$, where $\sigma(z) = 1/(1 + \exp(-x))$ is the sigmoid function. The invariance condition reduces to the same as for linear regression. We omit the details.

## C    More Examples on Augmented ERM

In this section, we give several examples of models where exact invariance occurs. We characterize how much efficiency we can gain by doing data augmentation and compare it with various other estimators. Some examples are simple enough to give a finite-sample characterization, whereas others are calculated according to the asymptotic theory developed in the previous section.

Figure 2: Plots of the increase in efficiency achieved by data augmentation in a *flip symmetry* model.

## C.1 Exponential families

We start with exponential families, which are a fundamental class of models in statistics [e.g., 46, 47]. Suppose $X \sim \mathbb{P}_\theta$ is distributed according to an exponential family, so that the log-likelihood can be written as

$$\ell_\theta(X) = \theta^\top T(X) - A(\theta),$$

where $T(X)$ is the sufficient statistic, $\theta$ is the natural parameter, $A(\theta)$ is the log-partition function. The densities of $\mathbb{P}_\theta$ are assumed to exist with respect to some common dominating $\sigma$-finite measure. Then the score function and the Fisher information is given by

$$\nabla \ell_\theta(X) = T(X) - \nabla A(\theta), \quad I_\theta = \text{Cov}\left[T(X)\right] = \nabla^2 A(\theta).$$

Given invariance with respect to a group $G$, by Theorem 3.5, the asymptotic covariance matrix of the augmented maximum likelihood estimator, $\hat{\theta}_{\text{aMLE}}$ (which is a special case of ERM with the loss function being the log-likelihood function), equals $I_\theta^{-1} J_\theta I_\theta^{-1}$, where $J_\theta$ is the covariance of the orbit-averaged sufficient statistic $J_\theta = \text{Cov}_X \mathbb{E}_g T(gX)$.

Moreover, the augmented MLE can be expressed as the solution of the following optimization problem, where we replace the sufficient statistic $T(x)$ by $\bar{T}(x) = \mathbb{E}_g T(gx)$:

$$\hat{\theta}_{\text{aMLE}} \in \arg\max_\theta \quad \theta^\top \mathbb{E}_g T(gX) - A(\theta).$$

We then have $\hat{\theta}_{\text{aMLE}} = [\nabla A]^{-1} \mathbb{E}_g T(gX)$.

An alternative strategy, which also exploits the invariance structure, is *constrained optimization*. Indeed, let $p_\theta$ be the density of $X$. If we assume the action of $G$ on the sample space is *linear*, then by a change of variable formula, the invariance relation $gX =_d X$ is equivalent to the following equation:

$$p_\theta(x) = p_\theta(g^{-1}x)/|\det(g)|.$$

The above equation translates to the following constraint on $\theta$:

$$\ell_\theta(gx) + \log|\det(g)| = \ell_\theta(x) \ \forall g \in G, x \in \mathcal{X}.$$

Let $\Theta_G \subseteq \Theta$ be the *invariant subspace* of the original parameter space, consisting of all $\theta$'s that satisfy the above equation. For exponential families, $\Theta_G$ consists of all $\theta$'s s.t.

$$\theta^\top [T(gx) - T(x)] + v(g) = 0 \ \forall g \in G, x \in \mathcal{X}. \tag{14}$$

where $v(g) = \log|\det g|$ is the log-determinant. This is a set of linear equations in $\theta$. Moreover, the log-likelihood is concave, and hence, we can in principle compute the following *constrained maximum likelihood estimator*:

$$\hat{\theta}_{\text{cMLE}} \in \arg\max_\theta \quad \theta^\top T(X) - A(\theta)$$

$$s.t. \quad \theta^\top [T(gx) - T(x)] + v(g) = 0 \ \forall g \in G, x \in \mathcal{X}.$$

Assume that $\Theta = \mathbb{R}^p$, so that the exponential family is well defined for all natural parameters, and that $\nabla A$ is invertible on the range of $\mathbb{E}_g T(gX)$. The KKT conditions of the above convex program is given by

$$\hat{\theta}_{\text{cMLE}} \in [\nabla A]^{-1}(T(X) + span\{T(gz) - T(z) : z \in \mathbb{R}^d, g \in G\})$$
$$s.t. \quad \theta^\top [T(gx) - T(x)] + v(g) = 0 \quad \forall g \in G, x \in \mathcal{X}.$$

### C.1.1 Gaussian Mean Estimation

Consider now the important special case of Gaussian mean estimation. Suppose that $X$ is a standard Gaussian random variable, so that $A(\theta) = \|\theta\|^2/2$, and $T(x) = x$.

Assume for simplicity that $G$ acts *orthogonally*. Then the constraints in Equation (14) is simplified to $g^\top \theta = \theta$ for any $g \in G$. Recall that maximizing the Gaussian likelihood is equivalent to minimizing the distance $\|\theta - X\|_2^2$. Hence, the constrained MLE, by definition of the projection, takes the following form:

$$\hat{\theta}_{\text{cMLE}} = P_G X,$$

where we $P_G$ is the orthogonal projection operator onto the tangent space of $\Theta_G$ at $\theta$. However, since $\Theta_G$ is a linear space in our case, $P_G$ is simply the orthogonal projection operator onto $\Theta_G$. On the other hand, we have

$$\hat{\theta}_{\text{aMLE}} = \mathbb{E}_{g \sim \mathbb{Q}}[g]X.$$

In fact, under the current setup, the augmented MLE equals the constrained MLE:

**Proposition C.1.** *Assume $G$ acts linearly and orthogonally. If $X$ is $d$-dimensional standard Gaussian, then $P_G = \mathbb{E}_{g \sim \mathbb{Q}}[g]$, so that both the aMLE and cMLE are equal to the projection onto the invariant subspace $\Theta_G$. In particular, their risk equals $\dim \Theta_G$.*

*Proof.* Let $C = \mathbb{E}_{g \sim \mathbb{Q}}[g]$. By orthogonality, for each $g$ we have that $g^\top = g^{-1}$ is also in $G$. Hence, the matrix $C$ is symmetric. Then for any $v \in \Theta_G$, we have $Cv = \mathbb{E}_{g \sim \mathbb{Q}}[gv] = \mathbb{E}_{g \sim \mathbb{Q}}[v] = v$. Moreover, for any $w \in \Theta_G^\perp$, we have $Cw = \mathbb{E}_{g \sim \mathbb{Q}}[gw] = \mathbb{E}_{g \sim \mathbb{Q}}[0] = 0$. Hence, $C$ is exactly the orthogonal projection into the subspace $\Theta_G$, which finishes the proof. $\square$

For instance, suppose $G = \{1, -1\}$ is the reflection group (acting by multiplication). Then it is clear that $\Theta_G = \{0\}$, and so both the cMLE and aMLE are identically equal to zero.

### C.1.2 Some Numerical Results

We present some numerical results to support our theory for exponential family models.

In Figure 2, we show the results of two experiments. On the left figure, we show the histograms of the mean squared errors (normalized by dimension) of the MLE and the augmented MLE on a $d = 100$ dimensional Gaussian problem. We repeat the experiment $n_{MC} = 100$ times. We see that the MLE has average MSE roughly equal to unity, while the augmented MLE has average MSE roughly equal to one half. Thus, data augmentation reduces the MSE two-fold.

On the right figure, we change the model to each coordinate $X_i$ of $X$ being sampled independently as $X_i \sim Poisson(\lambda)$. We show that the relative efficiency (the relative decrease in MSE) of the MLE and the augmented MLE is roughly equal to two regardless of $\lambda$.

### C.2 Parametric Classification Models

Consider a random sample $\{(X_1, Y_1), ..., (X_n, Y_n)\} \subseteq \mathbb{R}^d \times \{0, 1\}$ from the law of a random vector $(X, Y)$, which follows the model:

$$\mathbb{P}(Y = 1 \mid X) = \sigma(f(\theta_0, X)),$$

where $\theta_0 \in \mathbb{R}^p$. Here, $\sigma : \mathbb{R} \to [0, 1]$ is an increasing activation function, and $f(\theta_0, \cdot)$ is a real-valued function. For example, the sigmoid $\sigma(x) = 1/(1 + e^{-x})$ gives the logistic regression model, using features extracted by $f(\theta_0, \cdot)$.

We have a group $G$ acting on $\mathbb{R}^d \times \{0, 1\}$ via

$$g(X, Y) = (gX, Y),$$

and the invariance is
$$(gX, Y) =_d (X, Y).$$
The interpretation of the invariance relation is two-fold. On the one hand, we have $gX =_d X$. On the other hand, for almost every (w.r.t. the law of $X$) $x$, we have
$$\mathbb{P}(Y = 1 \mid gX = x) = \mathbb{P}(Y = 1 \mid X = x).$$
The LHS is $\sigma(f(\theta_0, g^{-1}x))$, whereas the RHS is $\sigma(f(\theta_0, x))$. This shows that for any (non-random) $g \in G$ and $x$, we have
$$\sigma(f(\theta_0, gx)) = \sigma(f(\theta_0, x)).$$
For image classification, the invariance relation says that the class probabilities stay the same if we transform the image by the group action. Moreover, since we assume $\sigma$ is monotonically strictly increasing, applying its inverse gives
$$f(\theta_0, gx) = f(\theta_0, x).$$

We consider using the least square loss to train the classifier:
$$L(\theta, X, Y) = (Y - \sigma(f(\theta, X)))^2.$$
Though this is not the most popular loss, in some cases it can be empirically superior to the default choices, e.g., logistic loss and hinge loss [72, 55]. The loss function has a bias-variance decomposition:
$$\mathbb{E}L(\theta, X, Y)$$
$$= \mathbb{E}[Y - \sigma(f(\theta_0, X)) + \sigma(f(\theta_0, X)) - \sigma(f(\theta, X))]^2$$
$$= \underbrace{\mathbb{E}[Y - \sigma(f(\theta_0, X))]^2}_{\mathbb{E}L(\theta_0, X, Y)} + \mathbb{E}[\sigma(f(\theta_0, X)) - \sigma(f(\theta, X))]^2,$$
where the cross-term vanishes because $\sigma(f(\theta_0, X)) = \mathbb{E}[Y|X]$. Note that
$$\mathbb{E}[Y - \sigma(f(\theta_0, X))]^2 = \mathbb{E}[(Y - \mathbb{E}[Y|X])^2]$$
$$= \mathbb{E}\left[\mathbb{E}[(Y - \mathbb{E}[Y|X])^2 \mid X]\right]$$
$$= \mathbb{E}\text{Var}(Y|X)$$
$$= \mathbb{E}\left[\text{Var}\left(\text{Bernoulli}(\sigma(f(\theta_0, X)))\right)\right]$$
$$= \mathbb{E}\sigma(f(\theta_0, X))(1 - \sigma(f(\theta_0, X))).$$
Meanwhile, since $\nabla\sigma(f(\theta, X)) = \sigma'(f(\theta, X))\nabla f(\theta, X)$, for sufficiently smooth $\sigma$, we have a second-order expansion of the population risk:
$$\mathbb{E}L(\theta, X, Y) = \mathbb{E}L(\theta_0, X, Y) + \frac{1}{2}(\theta - \theta_0)^\top \mathbb{E}[2\sigma'(f(\theta_0, X))^2 \nabla f(\theta_0, X)\nabla f(\theta_0, X)^\top](\theta - \theta_0)$$
$$+ o(\|\theta - \theta_0\|^2).$$
This suggests that we can apply Theorem 3.5 with
$$V_{\theta_0} = \mathbb{E}[2\sigma'(f(\theta_0, X))^2 \nabla f(\theta_0, X)\nabla f(\theta_0, X)^\top]$$
and
$$\nabla L(\theta, X, Y) = -2(Y - \sigma(f(\theta, X)))\sigma'(f(\theta, X))\nabla f(\theta, X),$$
which gives
$$\sqrt{n}(\hat{\theta}_n - \theta_0) \Rightarrow \mathcal{N}(0, \Sigma_0),$$
where the asymptotic covariance is
$$\Sigma_0 = \mathbb{E}[U_{\theta_0}(X)]^{-1}\mathbb{E}[v_{\theta_0}(X)U_{\theta_0}(X)]\mathbb{E}[U_{\theta_0}(X)]^{-1}$$
$$v_{\theta_0}(X) = \sigma(f(\theta_0, X)) \cdot (1 - \sigma(f(\theta_0, X)))$$
$$U_{\theta_0}(X) = \sigma'(f(\theta_0, X))^2 \nabla f(\theta_0, X)\nabla f(\theta_0, X)^\top.$$

Here $v_{\theta_0}(X)$ can be viewed as the noise level, which corresponds $\mathbb{E}\varepsilon^2$ in the regression case. Also, $U_{\theta_0}(X)$ is the information, which corresponds to $\mathbb{E}\nabla f(\theta_0, X)\nabla f(\theta_0, X)^\top$ in the regression case. The classification problem is a bit more involved, because the noise and the information do not decouple (they both depend on $X$). In a sense, the asymptotics of classification correspond to a regression problem with heteroskedastic noise, whose variance depends on the mean signal level.

In contrast, applying Theorem 3.5 for the augmented loss gives

$$\sqrt{n}(\hat{\theta}_{n,G} - \theta_0) \Rightarrow \mathcal{N}(0, \Sigma_G),$$

where

$$\Sigma_0 - \Sigma_G = V_{\theta_0}^{-1}\mathbb{E}\text{Cov}_G\nabla L(\theta_0, gX)V_{\theta_0}^{-1}.$$

We now compute the gain in efficiency:

$$\mathbb{E}\text{Cov}_g\nabla L(\theta_0, gX) = \mathbb{E}\text{Cov}_g\left(2(Y - \sigma(f(\theta_0, gX)))\sigma'(f(\theta_0, gX))\nabla f(\theta_0, gX)\right)$$

$$= 4\mathbb{E}\left[(Y - \sigma(f(\theta_0, X)))^2\text{Cov}_g\left(\sigma'(f(\theta_0, gX))\nabla f(\theta_0, gX)\right)\right]$$

$$= 4\mathbb{E}\left[v_{\theta_0}(X)\text{Cov}_g\left(\sigma'(f(\theta_0, gX))\nabla f(\theta_0, gX)\right)\right].$$

In summary, the covariance of ERM is larger than the covariance of augmented ERM by

$$\Sigma_0 - \Sigma_G = \mathbb{E}[U_{\theta_0}(X)]^{-1} \cdot \mathbb{E}\left[v_{\theta_0}(X)\text{Cov}_g\left(\sigma'(f(\theta_0, gX))\nabla f(\theta_0, gX)\right)\right] \cdot \mathbb{E}[U_{\theta_0}(X)]^{-1}.$$

In fact, one can get analogous results for under-parameterized two-layer nets, like those in Theorem 4.2. Since the calculations are essentially identical to the proof of Theorem 4.2, we omit the details.

# D  Experiment Details

Our experiment to generate Figure 1(a) (from the main paper) is standard: We train ResNet18 [33] on CIFAR10 [43] for 200 epochs, based on the code of `https://github.com/kuangliu/pytorch-cifar`. The CIFAR10 dataset is standard and can be downloaded from `https://www.cs.toronto.edu/~kriz/cifar.html`. We use the default settings from that code, including the SGD optimizer with a learning rate of 0.1, momentum 0.9, weight decay $5 \cdot 10^{-4}$, and batch size of 128. We train three models: (1) without data augmentation, (2) horizontally flipping the image with 0.5 probability, and (3) a composition of randomly cropping a $32 \times 32$ portion of the image and random horizontal flip; besides the data augmentation, all other hyperparameters and settings are kept the same. We repeat this experiment 15 times and plot the average test accuracy for each training epoch. This experiment was done on a p3.2xlarge (GPU) instance on Amazon Web Services (AWS).