[Reviews · NeurIPS 2020]

Review 1

Summary and Contributions: After Author's Response: After reviewing the authors' response, I maintain my overall score for the paper. Additional comments are added below. ========================================================= This paper studies data augmentation from a group theoretic perspective. Given that the actions of the data-augmentation forms a group, the authors show that stochastic gradient descent (SGD) with stochastic application of data-augmentation can be viewed as SGD on a group-averaged loss. The authors then prove that the "improvements" (of the loss and MSE of the discovered weights) that come from using this group-averaged training loss is bounded above by a quantity that is expressed as a sum of a variance reduction term and a bias term (which vanishes when the dataset is exactly invariant under the group action.) Some toy examples (2-layer networks, the circular shift model) are used to demonstrate the implications of these bounds.

Strengths: The authors provide rigorous statements on the effect of data-augmentation by defining metrics to measure improvement, and proving upper-bounds on these metrics. The bounds have an interesting decomposition into a variance reduction term and a bias term, which can potentially be used as a measure for assessing "augmentation quality." These contributions provide a new perspective on data augmentation.

Weaknesses: After Author's Response: I would like to reiterate the view that the authors seemed to have let some promising opportunities go by in terms of exploring the practical implications of their work. For example, I am curious how the relative efficiency of Theorem 4.1. correlates with improvements in performance coming from an augmentation. (It would have been very impressive if the authors were able to demonstrate something like this.) To comment on the author's response to estimating the variance reduction term---I think using trained weights as a proxy for theta_0 is a valid approach. The authors comment that training has been used to obtain theta_0, but that's okay. The goal would be to predict the effect of augmentation without training with augmentation. Training without augmentation to get theta_0 should be considered fair. ============================================================ There are two main issues that are worth pointing out. First, the paper has limited experimental verification of their results. In particular, only very toy models are used for experimentation. While it is understood that the assumptions made for many of the results are quite strict and rule out more complicated tasks and networks, it would have been desirable to have some sense of how these bounds are manifested in networks and tasks that are less-toy. For example, even experimentation on something like Fashion-MNIST on 2-layer networks with flip would have been interesting to see. Second, the bounds and results provided seem to have ripe potential for suggesting ways to judge the quality of an augmentation without actual training. While the authors may feel that such investigation is beyond the scope of the paper, it would have been nice to see the authors elaborate/expand on this point further in the paper (some hints of this sort of discussion can be found in section 4) or discuss how to compute various components (variance reduction, bias) that go into the bound, and estimate how expensive that would be compared to actually training the network with augmentation. In fact, it would be good to know how if the authors have ideas on how to estimate the variance reduction terms in more realistic settings. This is especially relevant, since the utility of inequalities often come from bounding a quantity that is hard to compute by something that is easier to compute.

Correctness: I am "fairly confident" the claims are correct. The paper does not focus on experimentation.

Clarity: The paper is could be a little more reader friendly, although this reviewer sympathizes with the author's predicament of not having enough space to able to do so.

Relation to Prior Work: Yes.

Reproducibility: Yes

Additional Feedback: There are a couple minor improvements this reviewer would like to suggest: 1. In Proposition 3.3 (line 145), \theta_0 is introduced without definition. It is defined later on at line 164, but being introduced that this point without definition can be confusing, especially because of the notation \theta_t introduced in equation (2) to mean the parameter at time step t of SGD. 2. The statement from line 148-149 seems to imply that \mathbb{E}_X \tr (\Cov_g \hat{\theta}(gX)) = Var(\hat{\theta}_G) - Var(\hat{\theta}). Should there be a minus sign on the left hand side, i.e,. shouldn't the first term on the RHS of (4) (with the minus sign present) be equal to Var(\hat{\theta}_G) - Var(\hat{\theta})?


Review 2

Summary and Contributions: The paper is very well written and constitute the first (to my knowledge) broad theoretical rigorous analysis of the effect of data augmentation in learning. I enjoyed reading it and I definitely recommend for publication.

Strengths: The paper is a novel theoretically sound mathematical framework for the analysis of the effects of data augmentation. This is very relevant to the community since data augmentation is a common method for improving accuracy in NNs .

Weaknesses: One of the most important dataset feature for their derivation is it invariance w.r.t a group transformations which I think boils down for the dataset to be a collection of orbits with respect to a group. Two related questions: -Real word data transformations are not group transformations, but locally, if the transformation is smooth enough the manifold associated to an object transformation locally has group-lie generators. Could the authors reason how their results could be changed to include this scenario? Could "local" data augmentation be reflected into a "local" loss average? -Suppose the dataset is composed by defective orbits i.e. not complete but missing some object transformations. Can the authors speculate how to use concentration type inequalities to change their bounds to accommodate for this specific form of noise?

Correctness: Yes, to my knowledge.

Clarity: The paper is very well written.

Relation to Prior Work: Yes, the authors discuss previous work and clarify their novel contribution.

Reproducibility: Yes

Additional Feedback: After reading the authors feedback I am satisfied with their replies and still think this is an excellent paper! Absolutely recommend for publication.


Review 3

Summary and Contributions: EDIT: After reading the author feedback and other reviews, I have opted to maintain my score as is. I believe the authors have set up a wide-ranging, novel framework for understanding data augmentation on a statistical level. Moreover, given the preponderance of empirical evidence on the efficacy of data augmentation in neural networks, I do not think that this paper needs to duplicate these experiments. I am happy that the authors instead opted to use their time and page limit to giving intuitive and complete discussions of their work. ----------------- This paper considers the benefits of data augmentation in statistical estimation. Notably, the paper analyzes a general class of group-theoretic data augmentation strategies (in which the group action leaves the data distribution approximately invariant) that capture many realistic data augmentation strategies (such as image rotations and flips). The authors show that when the distribution is invariant under the group, data augmentation necessarily leads to reduced variance in statistical estimation. When the distribution is only approximately invariant, the authors show a novel "invariance-variance" tradeoff that uses techniques from optimal transport. Finally, the authors apply this variance/invariance framework to Neural Tangent Kernels to show the benefits of data augmentation in training two layer neural networks.

Strengths: The work in question has many strengths. The paper is extremely sound (as far as I can tell), and derives a whole host of novel results. More generally, the entire framework of the paper is novel; As the authors note, most previous studies of data augmentation are primarily quantitative. While there are some that go beyond this, Paper 1260 is the most rigorous and far-reaching paper I have seen on the topic. In addition to novelty, I believe the authors' results have a large amount of potential for understanding and designing new data augmentation methods. The fact that the authors are able to derive variance-reduction results for such a wide class of estimation and data augmentation problems suggests that there may be specializations that prove enormously helpful. In addition, the fact that the authors then are actually able to show how their work manifests in realistic scenarios (ie. two-layer overparameterized networks) is quite impressive. Even without Section 3.4, I believe this paper has more than enough novel results to warrant acceptance. Finally, the invariance-variance trade-off seems to me to not just be a useful quantitative result, but a useful general framework for thinking about data augmentation strategies. I believe it is one of the highlights of the paper, and has the potential to inspire work that dives deeper into this topic.

Weaknesses: The primary weakness of the work is that the authors clearly have more to say on the topic than they can fit into 8 pages. As a result, the paper seems to come to an abrupt halt at the end of Section 4.1. However, given the scope and "meta" viewpoint of the paper (that data augmentation can be considered through an orbit averaging lens), I think that a discussion that not only summarizes but synthesizes the various results in the paper would be hugely helpful. In particular, the section on the circular shift model, while interesting to the reader who has entirely bought into the paper, seems somewhat extraneous to the core of the paper. It may be beneficial to either trim the discussion of this section (for the sake of some kind of summary discussion) or else provide greater motivation for the circular shift model. In particular, it seems like a similar framework could be used to consider translation of images in convolutional networks, which would certainly be of interest. However, this advice does not factor into my score at all, as I believe that even as currently written, the paper is more than deserving of acceptance.

Correctness: While I have skimmed the appendix to look for any obvious errors, I have not verified all technical details (especially as the paper requires a good number of sophisticated techniques spanning statistical estimation and optimal transport). However, those proofs that I have looked at in detail have been very well-written and clear. I therefore believe that the paper is sound. My one issue with the empirical study is that Figure 1b may be meaningless to a reader until they have fully digested Section 4. Instead, it may be worth presenting a well-known metric that can be linked (even heuristically) to relative efficiency. For example, one could compute accuracy of the two-layer network in Section 4.1 trained on a Gaussian input (where the label is y is given by evaluating the network on some circulant matrix W^* determined a priori), both with and without circular data augmentation. This could be done for a few values of d, and ideally we could see that the benefit of data augmentation scales with d. This suggestion above is clearly just this reviewer's preference. A simpler fix would be to introduce an intuitive notion of "relative efficiency" before presenting the plot. However, I do believe that something should be done to make Figure 1b have impact on the reader.

Clarity: One of the paper's greatest strengths is how well-written it is. The authors guide the reader through the theory, motivating it with conceptual ideas and discussing the theoretical underpinnings of this work. It was frankly enjoyable to read, and I believe it does a tremendous job in making some of the more abstract results interpretable to the layman.

Relation to Prior Work: The work does a great job of framing its contribution in relation to other works. The discussion of invariant architectures (for instance) is well done and helps the reader see the utility of this paper. My only issue here is that the few works that study data augmentation in a non-qualitative manner are mentioned only in passing. If anything, the discussion of these works in a slightly more extended way could server the paper well by emphasizing just how significant a jump in theory this current work provides.

Reproducibility: Yes

Additional Feedback:

[Author Response · NeurIPS 2020]

We would like to thank all the reviewers for their detailed and enlightening reviews, especially during these challenging
times.

**Response to Reviewer #1**

*Limited experimental verification.* Thank you for pointing this out. When we were preparing this manuscript, we were
having a hard time deciding how many empirical evaluations to include. After some thoughts, we finally decided to
only include the current Fig. 1 for the following reasons: (1) since this paper is primarily on the theoretical side and in
view of the 8-page limit, we prefer to use the limited spaces to explain the intuitions behind our technical results; (2)
the empirical success of data augmentation is already established in many other works, so we feel that presenting a
comprehensive empirical study may be of secondary importance in this paper.

We will do our best to do the Fashion-MNIST experiment with left/right flip. We need to figure out how to estimate
the variance reduction term (see next answer), but we will think about this and do our best to address it, at least in the
special case of the left/right flip.

*Estimating the variance reduction term.* Good point! This is indeed a super interesting direction and we have been
thinking about it from the very beginning of this project. The major difficulty along this direction is that, to estimate
the asymptotic variance reduction (say, based on Eq. 10) requires knowledge about the ground truth parameter $\theta_0$,
which may be hard to obtain. Some ideas we have in mind include: (1) replacing $\theta_0$ with the trained weights, but
this requires training and violates the original goal of "judge the quality of an augmentation without training"; (2)
replacing $\theta_0$ with a random initialization, which may be accurate in the neural tangent kernel regime when the network
is very wide. Another related idea is to start with a "candidate augmentation" $g$, and estimate $D((gX, Y), (X, Y))$
from the data for some distance measure $D$ between probability distributions. Then the estimated $\hat{D}((gX, Y), (X, Y))$
can be taken as an estimate of "how invariant our data are w.r.t. $g$". More concretely, for example, we can sample
$\{(x_i, y_i)\}_{i=1}^m$ from our training data, apply $g$ to each of them, and calculate the Wasserstein distance between the two
empirical distributions (which can be solved by a linear program). Somewhat related to the above idea, we may adopt a
hypothesis testing framework and try to test $H_0 : D((gX, Y), (X, Y)) \leq \varepsilon$ v.s. $H_1 : D((gX, Y), (X, Y)) > \varepsilon$. We
have not experimented with these ideas, but we believe these are interesting future directions, and we plan to explore
them further in the future.

*Miscellaneous questions.* Sorry for the confusion on $\theta_0$. And the statement from line 148-149 indeed lacks a minus
sign. We will correct these two issues in the final version of this paper.

**Response to Reviewer #2**

*Beyond group transformations.* Thank you for bringing this up. Indeed, the group structure is lacking in many
"real-world" transformations used by practitioners. The reason that we work with a compact group $G$ is because we can
endow it with a Haar probability measure, so we do not have measure-theoretic complications when computing averages
over the group. However, we would like to point out that all of our current results would hold if $G$ is only a semi-group
(i.e., a set of transformations, not necessarily invertible), provided we can endow it with a uniform probability measure
(which holds, for example, when $G$ is discrete). We will include a discussion on this point in the final version of this
paper.

*Defective orbits.* It would be very interesting to characterize, in a quantitative fashion, the performance loss induced by
defective orbits compared to the full orbits. We will explore along this direction in future works.

**Response to Reviewer #3**

*Abrupt halt at the end of Sec. 4.1.* Thank you for the concrete suggestion (to trim some of the section on the circular
shift model in favor of having more space for a final discussion section) and sorry for the abrupt halt, which is a
compromise we made in view of the 8-page limit. We will include a discussion section in the final version of this paper,
also incorporating some of the feedback from Reviewer #2.

*Clarification on Fig. 1b.* Sorry for the ambiguity of the purpose of Fig. 1b. We will at least explicitly define relative
efficiency and provide some intuition for it in the caption of Fig. 1b in the final version of this paper to better explain
the purpose of this plot. The suggestion of training a two-layer net with Gaussian inputs seems very interesting, and we
will do our best to include it in the final version of this manuscript (perhaps replacing the current Fig. 1b).

*Other quantitative works on data augmentation.* Sorry, this is again a compromise of the 8-page limit. We will comment
more on other quantitative works in the final version of this paper.

[Meta-Review · NeurIPS 2020]

All reviewers agreed on the theoretical value of this paper, explaining the effects on data augmentation through group theory. The results are novel, and very relevant to one of the most widely techniques used along with SGD in practice. Some concerns were raised on the breadth of experimental validation, and expanding on the potential of the presented theory to better inform practice, that the authors should consider. I want to further add that in my opinion this is an excellent and very timely paper, and I am really looking forward to the ways it will impact the related literature.